# Abyssal hydrothermal alteration drives the evolution from simple alkanes to prebiotic molecular complexity

Quanyou Liu [1] ✉, Huiyuan Xu [1,2,3] ✉, Jiuyuan Wang [1], Dongya Zhu[2], Chi Zhang[3], Biqing Zhu[1], Yin Fu[3], Wang Zhang [3], Jiangtao Li [4], Di Zhu[5], Shili Liao[6], Chunhui Tao [6], Guanghui Yuan [7], Shang Xu[7], Huaiwei Ni[8], Fang Hao[7] & Zhijun Jin[1]

Abyssal hydrothermal vents are regarded as reactors for simple reduced carbon transforming into more complex forms of prebiotic organic chemistry. While the organic geochemical continuum and evolutionary transitions remain elusive, due to the intense hydrothermal alteration. We apply a metabolomics-inspired molecular fingerprinting strategy integrating mass spectral networking and hierarchical organization, to construct a molecular relatedness phylogenetic tree for vents from ultraslow-spreading Indian Ridge. Here we show that organic molecules from different vent fields and activity states share common molecular connection patterns. The observed progressive molecular evolution from alkanes through aromatics to complex heteroatom-bearing compounds reveals a systematic increase in molecular functionalization and polarity. This finding helps bridge the gap between simple reduced carbon and prebiotic molecular complexity, underscoring the role of hydrothermal systems in shaping life's essential feedstock on the primordial Earth. This framework may contribute to the search for life-markers on other astrobiological contexts, e.g., Mars, Enceladus, Callisto and Europa.

Abyssal hydrothermal vents are recognized as potential reactors for prebiotic organic synthesis and molecular concentration in compartments[1–3], as they concentrate reduced carbon, metals and chemical energy, making them prime candidates for hosting the earliest steps of prebiotic chemistry. However, if ultramafic-hosted hydrothermal systems facilitated the abiotic synthesis of organic building blocks through initial processes such as serpentinization and $H_2$-$CO_2$ redox coupling, including methane, chained alkane, formate, sugar, ammonia and other simple organics essential for life's origin[4–8], and provided suitable conditions for their concentration and preservation[1,3,4], how can we reconcile the limited evidence for evolution pathways from the simple abiotic reduced alkanes (e.g., $C_xH_{2x}$) to the relatively complex building blocks (e.g., amino acids, nucleobases)[5,8]. How did the dynamic redox gradients and hydrothermal reactions in these environments specifically start to drive the selective accumulation and eventual evolution to hetero-bearing organic molecules as precursors for building blocks and proto-biogeochemical networks? What is the mechanism that favors the retention and assembly of labile prebiotic intermediate molecules over fluctuating thermal and chemical conditions under extreme

[1]Institute of Energy, School of Earth and Space Sciences, Peking University, Beijing, China. [2]Petroleum Exploration & Production Research Institute, Beijing, China. [3]Institute of Geology and Geophysics, Chinese Academy of Sciences, Beijing, China. [4]School of Ocean and Earth Science, Tongji University, Shanghai, China. [5]School of Sustainable Energy and Resources, Nanjing University, Suzhou, China. [6]State Key Laboratory of Submarine Geoscience, Second Institute of Oceanography, Ministry of Natural Resources, Hangzhou, China. [7]School of Geosciences, China University of Petroleum (East China), Qingdao, China. [8]School of Earth and Space Sciences, University of Science and Technology of China, Hefei, China. ✉e-mail: liuqy@pku.edu.cn; xuhuiyuan.ian@hotmail.com

environments[5,9,10]? Ultimately, what fundamental roles the abiotic reduced alkanes played as the most initial stimulators[8,11–17] for the hypothesis that life emerged from such settings[1].

The compositions of vent smokers reflect interactions among mantle-derived fluids, seawater, organics, etc. Hydrothermal circulation, along mid-ocean ridges, serves as critical analogs for understanding redox-driven organic synthesis and element cycling in extreme environments, and plays an important role in global ocean cycles via significant inputs of reduced substrates, such as $H_2S$, $H_2$, $CH_4$, $NH_3$, $Mn^{2+}$, $Fe^{2+}$, that fuel chemosynthetic metabolism[18,19], bridging lithospheric processes with sub-seafloor biospheres. The Indian Ridge, a critical node in global mid-ocean ridge systems[20], is an ultraslow-spreading ridge with widely distributed hydrothermal fields[21] (Fig. 1). It hosts one of Earth's most geochemically dynamic environments characterized by unique magmatic-tectonic interactions and complex organic-inorganic geochemical processes (see Supplementary geology). It's ultraslow spreading rate results in greater variability in magmatism (episodic supply), lithology, fluid circulation, and vent chemistry, influenced by detachment faults that expose lower-crustal and serpentinized mantle rocks[22]. Thus these hydrothermal systems vary in geological settings, basement rock compositions, and mineral associations[21], associated with massive sulfide-rich hydrothermal vents. Active hydrothermal vent fields in Indian Ridge, with vent fluid temperatures reaching ~380 °C, are rarely seen observed on ultraslow-spreading ridges compared to fast-spreading ones, making the Indian Ridge a key area for studying hydrothermal extrusion's effects on deep marine conditions[18], providing unique insights into life thriving under extreme conditions[18].

Despite abundant evidence for such abiotic products[7,8,11,13,15,23], the pathway gap that converts these simple reduced carbons into the heteroatom-bearing complex precursors necessary for prebiotic chemistry remains poorly constrained. Previous studies have been lacking about a continuous molecular record that tracks alteration from alkanes to polar, functionalized species and identifies the physicochemical drivers of that transition. The magnitude of mass spectral metadata, the complexity of hydrothermal organic assemblages, and the subtle trends and structural linkages manifested by untargeted compounds collectively expose the limitations of conventional targeted-compound protocols in organic geochemistry, such approaches cannot resolve, at the molecular scale, the systematic evolutionary pathways that remain a black-box process.

To help bridge this gap, and in light of the highly dynamic hydrothermal organic geochemistry in the context of mass spectral metadata and chemical ontologies in living seafloor systems, we applied a metabolomics-inspired strategy including high sensitivity and high resolution (hard and soft) ionization, deconvolution of mass spectra, high-precision structural identification and molecular similarity matching, and hierarchical organization of molecular fingerprints (predicted from untargeted fragmentation spectra), to construct a molecular relatedness tree (i.e., organic geochemical

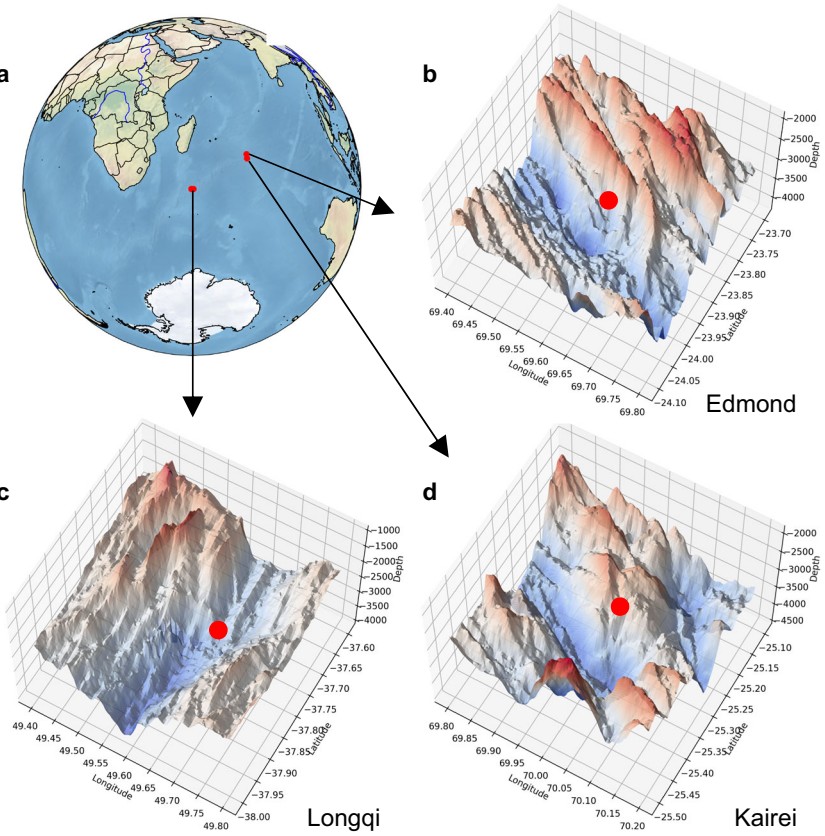

**Fig. 1 | Tectonic setting and high-resolution bathymetry of the Indian Ridge hydrothermal vent fields. a** Orthographic projection of global relief centered on the Indian Ocean. Red circles locate the Edmond, Kairei, and Longqi vent fields along the ultraslow-spreading Indian Ridge. The open-source Python library Cartopy was used for geospatial visualization and mapping (https://cartopy.readthedocs.io/stable/copyright.html; https://doi.org/10.5281/zenodo.1182735; https://github.com/SciTools/cartopy/blob/main/LICENSE)[71]. **b**–**d** Shaded-relief, three-dimensional digital elevation models derived from shipboard multibeam sonar for the Edmond (69.6°E, 23.9°S), Longqi (49.7°E, 37.8° S), and Kairei (70.0° E,

25.3° S) axial valleys, respectively. Color gradients transition from red (shallow) to blue (deep) to emphasize relief; depth scale bars are in meters. Red dots mark the precise positions of the vents sampled for geochemical and mineralogical analyses. Bathymetric data were obtained from the GEBCO 2024 gridded dataset, which provides global, publicly available seafloor elevation data (https://www.gebco.net/data-products/gridded-bathymetry-data; GEBCO Compilation Group (2025) GEBCO 2025 Grid (doi: 10.5285/37c52e96-24ea-67ce-e063-7086abc05f29)). The software used is open-source and publicly available.

phylogenetic tree) from mass spectral molecular networking of hydrothermal vent smokers with varying activity levels across different regions.

By building this tree on both active and inactive hydrothermal vents from Longqi, Edmond, and Kairei vent sites along the Indian Ridge (Supplementary Fig. 1), the specific aim of this study is to determine whether hydrothermal processes impose a shared molecular trajectory from simple alkanes to complex, functionalized N-, S- and O-bearing molecules, and to visualize the pathways of hydrothermal organic alterations and transformations in such systems[24], and to explore their implications for prebiotic chemistry in early Earth environments. This work may offer insights into the evolution of life-related substances in primordial ocean world, and this framework may also contribute to the search for deceased life markers on Mars providing a conceptual model for identifying biosignatures in astrobiological contexts.

## Results and discussion

Our strategy comprises three integrated steps: (1) molecular fingerprinting using complementary ionizations; (2) deconvolution of mass spectra to construct high-accuracy molecular networks; (3) organic geochemical deciphering through an organic geochemical relatedness tree that map transformation pathways. The deconvolution was employed only during data pre-processing to separate chimeric EI spectra into compound-specific spectra[25]. This purifies each chromatographic peak, yielding a single, high-fidelity composite mass spectrum per molecule and thereby eliminating the subjectivity and uncertainty introduced by manual or empirical operations. Once the spectra are purified (Supplementary Fig. 2), all subsequent calculations (pairwise cosine similarity, distance matrices, hierarchical clustering, and geochemical phylogenetic reconstruction; Here, phylogenetic is used by analogy to denote similarity-based hierarchical organization of molecular fingerprints, not a biological phylogeny of organisms) rely on well-established, deterministic chemoinformatics and statistical routines. A triple check of spectral data similarity (20%, 50% and 70%) was used to cross-validate the reproducibility of the peak spotting, the alignment and identification of organic compounds. The molecular fingerprints of shared systematic patterns and networks reveal coherent molecular transformation pathways that are spread across varied vent conditions and geographical locations. Mercury isotope analysis confirms the predominantly mantle-derived origin of the hydrothermal vent samples, with detailed data and interpretation provided in the Supplementary information.

### Bidirectional reaction pathways and molecular structural complexity

In geological hydrothermal systems, organic compounds undergo significant transformations influenced by elevated temperatures and interactions with inorganic species and mineral surfaces, and studies of these environments reveal complex and substantial hydrothermal effects on organic matter[24,26,27]. While organic compounds tend to decompose into simpler species like $CO_2$, $CH_4$, $H_2$, and $H_2O$ at full thermodynamic equilibrium, kinetic barriers can prevent the system from reaching this state[6,28–30]. Instead, many organic reactions in hydrothermal settings reach a state of metastable thermodynamic equilibrium, where reversible reactions attain steady-state ratios[27,30–34]. This suggests that while not all possible reactions achieve equilibrium, certain reactions involving more reactive species become reversible, with their relative abundances governed by specific prevailing conditions.

Organic compound speciation in deep-sea hydrothermal vents is controlled by the environmental energy regime: temperature, $H_2$, $O_2$, pH and Eh gradients[35] produced by magma-mineral-water interactions, and are driven by reaction buffering, with metal minerals acting as catalytic or redox buffers (e.g., FMQ, PPM)[6,36].

These conditions result in metastable, reversible transformations that involve oxidation–reduction, dehydration–hydration, carboxylation–decarboxylation, cracking–condensation and other fundamental reactions. Consequently, molecular profiles (composition and abundance) vary with changing vent conditions, and e.g., $CH_4 \leftrightarrow CO_2$, alkane↔alkene↔alcohol, carboxylic acid↔alkane+$CO_2$, alkylation/dealkylation hydrocyclization/aromatization/dehydrocyclization, serpentinization, sulfidation, heteroatomization and metal complexation are routinely observed[29,35,37–42].

Experimental studies have also well demonstrated the reversibility of specific organic reactions under hydrothermal conditions, including the interconversion of alkanes and alkenes[28,29,34], as well as substitution and hydrolysis reactions among compounds like alcohols, amines, ethers, and aldehydes. These reversible reactions allow organic systems to approach metastable equilibrium, resulting in predictable activity ratios between certain compound pairs based on the prevailing temperature and fluid composition.

Organic compounds in hydrothermal systems originate from both biotic and abiotic sources[11,34], both subject to significant hydrothermal alteration. The particular presence of unresolved complex mixtures (UCMs) (Figs. 2a, b) serves as characteristic signatures of hydrothermal alteration[24,40,43]. This alteration involves a combination of complicated processes beyond simple thermal maturation or decomposition, including oxidation, dealkylation, aromatization, ring opening, sulfur and oxygen incorporation, and dehydrocyclization, etc[24,28,32,34]. The transformation of less stable and more stable forms at one site can leave a shift mark in the molecular gradient. The resulting rarely-seen concurrent molecular patterns from sterol through steranes to aromatic steranes and then heterocyclic aromatic compounds (Fig. 2c–f) may reflect a maturity gradient of molecular structures driven by varying hydrothermal influence[24,40,44,45]. This observable molecular structural gradient reflects a continuous hydrothermal imprint from relatively chained/branched alkanes and light cyclic compounds towards increasingly complex, higher-maturity aromatic/heterocyclic structures[34], reflecting continuous molecular restructuring under hydrothermal conditions, progressing from simpler, lower-molecular-weight compounds, to more complex, highly transformed structures.

Ultimately, over geological timescales, subjected to the substantial thermochemical alteration, these organic alteration and molecular transformation processes occurred and accumulated, thereby demolishing most of the biological information and imprinting distinctive molecular signatures within smokers (Figs. 2 and 3; Supplementary Movie 1), and the complexity of the resultant organic molecular structures significantly increases[24].

### Network-informed molecular framework tracks organic evolution

One major reason that prevents scientists from recovering more geologically long-preserved organic structures and its biological information from chimney interior is the high thermal maturation of organic molecules at the hotter end of the temperature gradient[24,40], and another is the intense, ongoing alteration and transformation these molecules experience under such conditions. In contrast to studies that center on highly informative but thermally labile biomolecules (e.g., nucleic acids, metabolites) in genomics and metabolomics, which catalog hereditary features of extant, unaltered viable organisms that has escaped hydrothermal overprinting, our investigation focuses on the more thermally recalcitrant fraction (hydrocarbons, condensed aromatics and heterocyclics) residing in hot fluids and chimney interior matrices. The latter preserves a cumulative record of the vent's dynamic hydrothermal geochemical history, which reveals much earlier and fuller profiles and more primordial imprints of organic transformation and evolutionary processes in vents, offering a more complete geochemical record rather than those from extant microbial life. For example, genomics provides a heritable

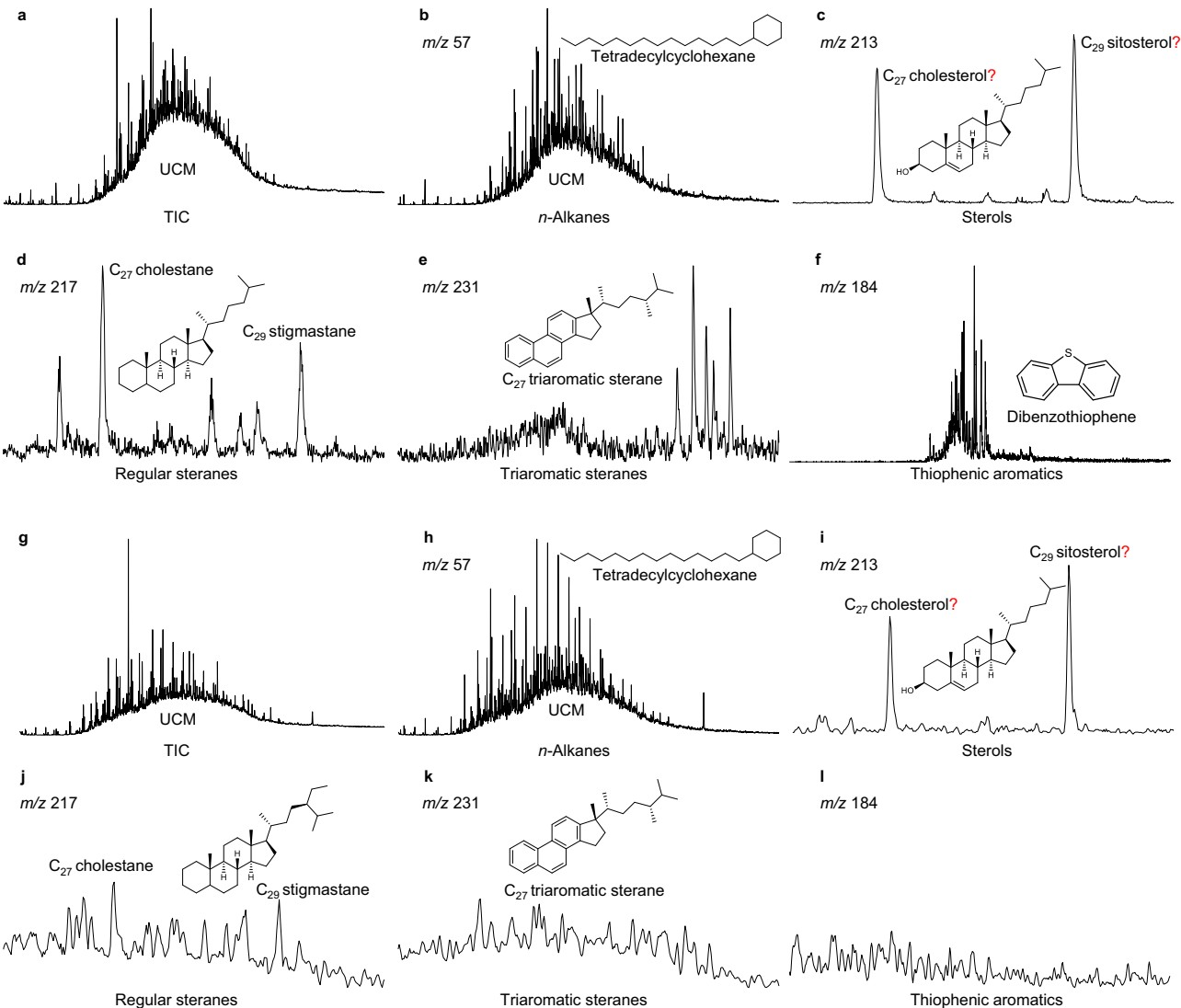

**Fig. 2 | Representative hard ionization mass chromatograms of organic compounds (including biomarkers) in Edmond vent SY139-G07 and Longqi vent SY107. a**, **b** Total-ion-current and *m/z* 57 mass chromatograms of the extract show a huge unresolved complex mixture (UCM). **b**–**f** Partial mass chromatograms (*m/z* 213, *m/z* 217, *m/z* 231 and *m/z* 184) show the diversity of commonly known hydrocarbons and biomarkers. An active site sample was shown in (**g**–**l**) (SY107; Longqi) for comparison. Skeletal structures mark the sterols, steranes, triaromatic steranes, S-bearing dibenzothiophene, attesting to thermally mature, and hydrothermally altered organic matter. These mass chromatograms capture a continuum from low-molecular-weight alkanes, through low maturity sterols and moderate maturity steranes, to high altered triromatic steranes and heterocyclic compounds within one vent sample in the ultraslow-spreading Indian Ridge vent system. The procedural blanks show no detectable sterol ions (see Supplementary Fig. 9), confirming that these trace level of sterols (with red question marks) are indigenous rather than laboratory contaminants.

framework through DNA sequences that record biological variation through time, and in an analogous way hydrothermal alteration and transformation of organic compounds provide a dynamic, process-based record of organic geochemical change driven by the hydrothermal environment. Rather than invoking biological phylogenetics, we adopt a perspective inspired by metabolomics, clustering chemically transformed compounds to explore their structural relationships and potential transformation pathways under environmental stress. Because these compounds possess markedly greater thermal stability, they preserve the dynamic imprint or gradient of hydrothermal alteration and transformation, offering a direct record of the reactor that vents provide for prebiotic molecular complexity.

The network-based approaches, originally developed for studying systematic molecular change in living systems[25], are adopted here to non-enzymatic hydrothermal systems, where thermodynamic constraints govern transformation networks (see Supplementary notes). This tracks the complex organic molecular transformations in such environments rather than focusing on individual biomarkers or organic compounds. Biological metabolomics where enzymes drive specificity, hydrothermal systems rely on thermochemical alteration. Both 37°C animal cells and 370°C abyssal hydrothermal black smokers show dynamic, reversible, multi-path networks of organic transformation that provide immediate feedback and regulation in response to external forcing in cells or hydrothermal environments. Just as metabolomics reveals metabolic pathways by analyzing molecular fingerprints and their systematic changes, we hypothesized that hydrothermal organic molecular complexity could be understood through similar network-based approaches, where molecular structural relationships reflect geochemical transformation pathways in the living abyssal hydrothermal systems. The resulting organic-geochemical spectrum captures both transient and cumulative geochemical fingerprints over different hydrothermal conditions. Network-topology models (such as the geochemical phylogenetic tree[25,46] in

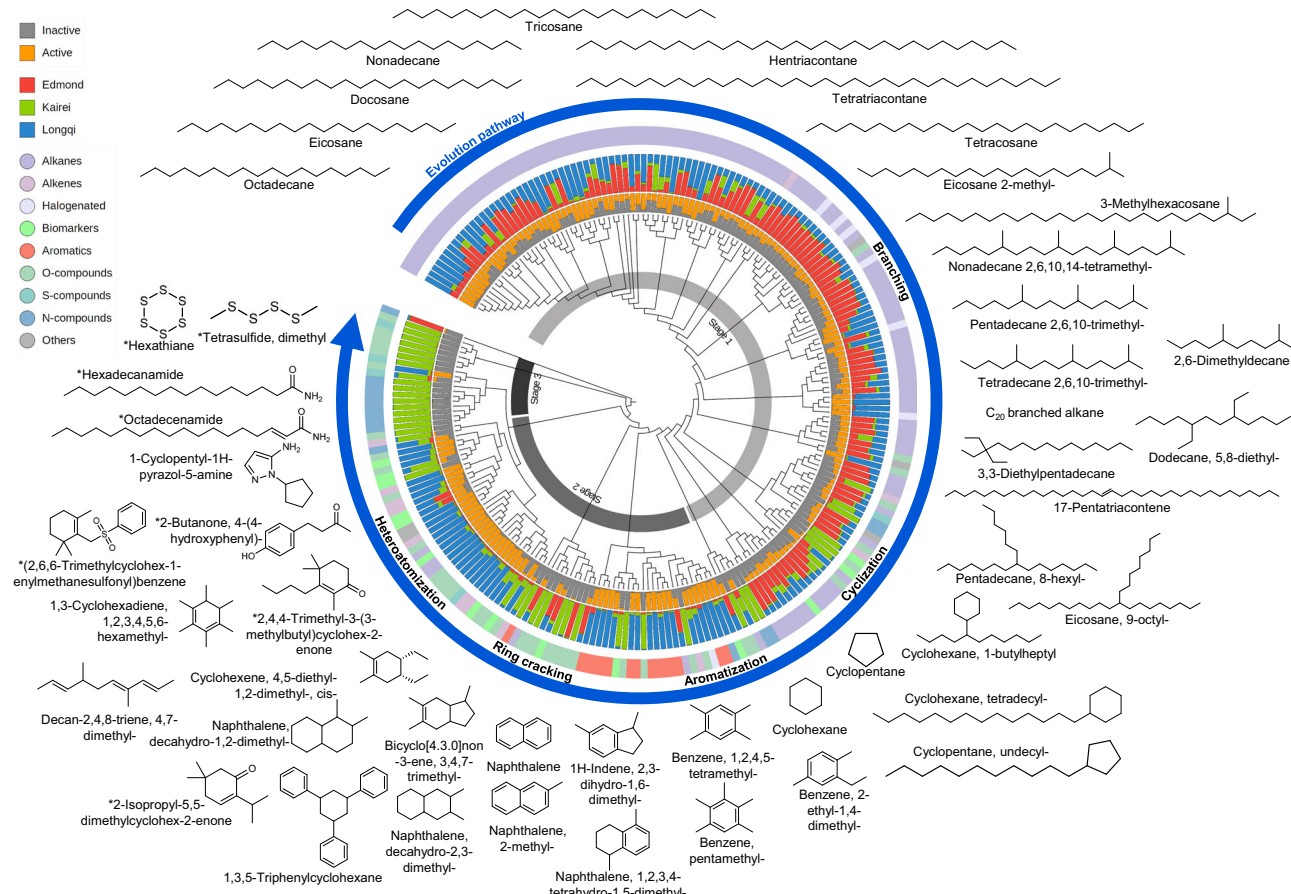

**Fig. 3 | A geochemical phylogenetic tree of organic compounds categorized by their chemical families across vent activities and locations.** The tree is based on predicted molecular fingerprints representing the structural relationships between compounds detected. The tree was generated using 233 representative organic compounds (out of 654 aligned organic compounds; from >213,000 similarity matches). The tree is organized into concentric bands, each representing a specific attribute. The innermost ring indicates three distinct transformation stages (gray gradient); the second innermost ring indicates vent activity status (inactive [gray] or active [orange]); the subsequent outer ring specifies the vent field origin (Edmond [red], Kairei [green], or Longqi [blue]). The outermost colored band highlights the chemical families of the organic compounds, including alkanes, alkenes, halogenated compounds, biomarkers, aromatics, oxygen-bearing compounds (O-compounds), sulfur-bearing compounds (S-compounds), nitrogen-bearing compounds (N-compounds), and other miscellaneous molecules. The distinct clades within the tree are grouped into three geochemical stages (Stage 1, Stage 2, and Stage 3), reflecting different phases of hydrothermal alteration and transformation succession. Representative skeletal structures and their names at each cluster, subclade and stage are provided based on spectral reference library (NIST20) matches obtained from feature-based molecular networking. Blue circle arrow and black texts indicate the presumed molecular evolution pathways and corresponding molecular transformations.

the context of various metadata and molecular annotations obtained from spectral matching[25,47–49]) reveal shared (thus a reduced scope of) molecular evolutionary pathways across vent settings. The geochemical phylogenetic tree is derived from comparative analysis of an extensive suite of organic-molecular fragment-ion signatures (e.g., Supplementary Fig. 3), with structural-similarity metrics quantifying relatedness among individual molecular fingerprints (Supplementary Dataset 1). Within the stepwise hydrothermal-alteration regime, assemblages of structurally similar molecules (branches or clusters on the tree) are interpreted to record common alteration stages or to share convergent transformation trajectories. The resulting continuum of similarity integrates these nodes into a coherent evolutionary pathway.

Although the precursor organic matter in this scenario should include biotic inputs, derived from microbial communities indigenous to or entrained within the hydrothermal system, a central premise is that the intense hydrothermal conditions have largely erased primary biological signatures. Instead, what remains is a set of highly transformed organic compounds that no longer retain bio-diagnostic features, but instead record the thermochemical legacy of the vent environment. Although some structures such as sterols

are thermally labile, their occurrence here likely reflects entrained biological inputs that underwent hydrothermal overprinting, importantly, contamination during extraction can be excluded based on the absence in blanks. In this context, the specific origin of a (biotic/abiotic) molecule becomes less critical than its role as a common organic substrate for the natural laboratory of the vent system, cycloalkanes, polycyclic aromatic hydrocarbons (PAHs) and thiophenes, etc., for example, persist not as diagnostic biomarkers but as a pool of chemically robust substrates continually reworked in the hydrothermal reactor, now essentially divorced from their original biotic/abiotic origins and transformed into molecular manifestations of hydrothermal processing. Vent samples from different regions with varying activity levels along the Indian Ridge could in principle have contained their own total molecular composition, however they most likely share certain proportions of common molecular geochemical characteristics. Thus those presumed shared characteristics and geochemical phylogenetic tree can provide essential constraints and critical contextual information for investigating mechanisms controlling organic molecular structural evolution and their metastable equilibrium states in abyssal extreme hydrothermal environments[34].

## Common evolution pathway from alkanes to nitric molecular complexity

We construct an organic geochemical relatedness tree, a network-based representation of molecular relationships inferred from untargeted mass-spectral fingerprints[46], analogous to phylogenetic tree for metabolites[46] or genomic sequences[50]. This usage is phylogeny-inspired for organizing molecular similarity, and does not imply biological ancestry. The clustering tree, comprising subclades, tree branches, or leaves, organizes organic molecules based on structural similarity inferred from molecular fingerprints (untargeted fragmented mass spectra), and each terminal node or tip represents a feature or compound, which enables synchronized chemically- and molecularly-informed comparisons of heterogeneous samples with varying vent activity levels across different regions in the ultraslow-spreading Indian Ridge (Fig. 3). The entire tree and its subclades, which reflect their inherent chemical structural relationship, are used to identify the hydrothermal effects on their shared changes in molecular structures and compositions, ultimately shedding lights on the unified pathway of organic molecular evolution under varying abyssal hydrothermal conditions.

From all vent rock extracts, 654 organic molecular structures were co-detected and co-annotated ('co-' indicates that the same compound mass spectrum were detected and annotated in all different samples) from every vent sample using hard ionization, and 233 organic compounds were identified with spectral match rate > 70%, as classified by alkanes, alkenes, halogenated-compounds, biomarkers (steroids), aromatics (i.e., PAHs and phenyls), NSO-compounds (amides, amine, thiophenes, furans, ester, ketone, acid, etc.) and others (Fig. 3; and Supplementary Fig. 3). Figure 3 provides geochemical phylogenetic relationships among the co-identified compounds, with each compound class clustered (outer ring of bars). The bar plots show the proportion of specific compounds over activities (inner ring of bars) and vent areas (middle ring of bars). This geochemical tree reveals three distinct stages (three bands within the tree; Stage 1, Stage 2, and Stage 3): 1) subclades of alkanes are mostly co-annotated in both active and inactive vents mostly from Longqi and Edmond regions; 2) subclades of aromatics and a mix subclades of alkenes, biomarkers and SO-compounds are mostly co-annotated in the active vents mostly from Longqi region; 3) in contrast, the subclades of NSO-compounds are extensively distributed in inactive vents mostly from the Kairei region.

The distribution of these compound classes shifts systematically clockwise around the tree. The lowermost subclades (Stage 1) are predominantly occupied by alkanes (purple), accompanied by a minimal proportion of biomarkers (light green). Straight-chain and branched alkanes appear in several smaller clusters within Stage 1, suggesting that the subclades of alkanes successfully groups with shared structural characteristics, and there is a clear clockwise branch-and cyclic-trending for alkanes at late Stage 1 (Fig. 3), which imply related origins or transformation pathways. At Stage 2 the middle branches, a clear transition emerges: chained/branched/cyclic alkanes diminish, while aromatics (PAHs and phenyls; salmon-red) increase markedly, notably in the active vents, suggesting a clear aromatization process. Going forward, hydrothermal ring cracking generates unsaturated alkenyl fragments that provide reactive handles for subsequent functionalization, and there is a pronounced rise in SO-compounds (mint-green, sea-green), with presence of alkenes and biomarkers, in mostly active vents. This suggests that ongoing hydrothermal fluid and high redox flux promote secondary alterations and transformations (e.g., ring opening, hydrogenation, etc.) of PAHs, alkenes and complex OS-bearing molecules. Clockwise from late Stage 2 to Stage 3, significant heteroatom incorporation (N, S, O) yields esters, ketones, carboxylic acids, thiophenes, amines, and amides. In Stage 3 the uppermost branches of the tree, there is a clearer enrichment in N-compounds (teal-blue), accompanied by SO-compounds.

Other features include the sporadic occurrence of halogenated compounds in certain branches, which may record pulses of high-halide flux or transient redox swings. This clear clockwise progression in molecular transformation of chained alkanes through branching, cyclization, aromatization, cracking and heteroatomization reflects increased molecular diversity and complexity and environmental interactivity (Fig. 3).

These organic geochemical phylogenetic distribution patterns can represent an evolution from simple reduced chained/branched hydrocarbons (possibly from an initial carbon pool by abiotic reduction[17] of $CO_2/H_2$ in high-temperature, metal-catalyzed settings), through PAHs and phenyl compounds (from cyclization, dehydrogenation, aromatization, etc. under elevated temperatures or shifting redox conditions), to complex N-, S- and O-bearing compounds (from a highly processed organic pool by condensation, sulfurization, amination, etc. under relatively cooler and more oxidizing plume environments). This implies that vent shut-down drives heterocyclic and especially nitric molecular enrichment, which aligns with the results of molecular formulae obtained by soft ionization (non-fragmented) mass spectrometry (Fig. 4).

This observed evolution pathway from chained/branched alkanes through aromatics to heteroatom-rich species, reflects a progressive molecular heteroatomization trajectory that is plausibly driven by the combined thermal and geochemical influence within hydrothermal vent systems. The shared compositional patterns observed in heterogeneous samples from vents of different activity levels and regions are best understood as aggregate responses to multiple environmental variables (e.g., heat, $O_2$, $H_2$, pH, redox potential, metals), rather than the effect of any single factor, that our current dataset cannot isolate the specific contributions of individual variables. In this sense, the integrated pattern provides observational constraints on how hydrothermal environments collectively shape organic molecular evolution, offering insights into the functional and structural transformation and dynamic metastability of organic molecules in abyssal hydrothermal contexts. Future work involving more resolved geochemical measurements and controlled sampling strategies will be essential to disentangle the relative influence of key environmental drivers.

Despite that the observed unified pathways or chains may reflect the progressive degradation of pristine organic matter generating increasingly lower molecular weight products over thermal alteration processes, it is essential to also consider the complex equilibrium dynamics operating within hydrothermal environments. Under hydrothermal conditions, organic compounds exist in metastable equilibrium states where both forward and reverse reactions occur simultaneously. This bidirectional reaction potential creates a dynamic pathway system in which simple compounds can recombine to form more complex structures, while concurrent degradative pathways continue to generate simpler molecular components. The interplay between these opposing reaction trajectories may give rise to a complex transition network that is likely influenced by the overall physicochemical conditions of hydrothermal systems. This metastable equilibrium framework provides a more nuanced interpretation of the observed molecular distributions and the overall compositional trends, suggesting that hydrothermal organic geochemistry functions as a dynamic, reversible system rather than a simple, unidirectional degradation sequence.

A continuum of related compounds has been reported from fossilized crustacean and mineral deposits[40,51], where it can be attributed to microbial transformation and sedimentary diagenesis and thermochemical alteration. A molecular continuum may arise from multiple processes including diagenetic transformation, microbial alteration, thermal stress, and hydrothermal overprinting. However, the microbial mechanisms are unlikely to dominate in this scenario. The vent interiors examined here experience rapid hydrothermal heating (> 300 °C), low TOC, and steep redox and temperature gradients,

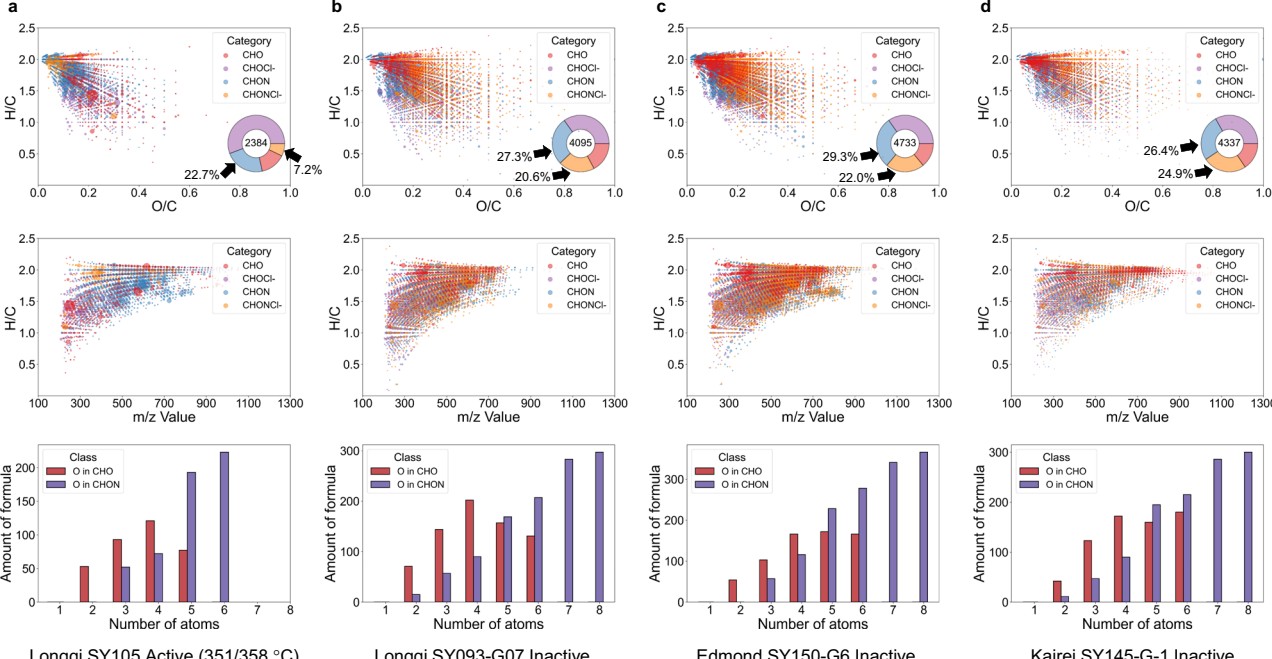

Longqi SY105 Active (351/358 °C)   Longqi SY093-G07 Inactive   Edmond SY150-G6 Inactive   Kairei SY145-G-1 Inactive

**Fig. 4 | Fourier transform mass spectra data in electrospray ionization mode of the extracts from vent samples in Longqi, compared with those in Edmond and Kairei. a–d** Data visualization of the chemical compositions and number of molecules, including the van Krevelen diagrams (H/C versus O/C and *m/z* value [from 100 to 1300]) and the number of molecular formulae as a function of number of oxygen atoms in the CHO, and CHNO chemical families for molecular formulae in Longqi (active, **a** and inactive, **b**), Edmond (inactive, **c**) and Kairei (inactive, **d**).

Individual bubbles represent formula and are colored by specific chemical families: CHO (red), CHOCl- (purple), CHON (light-blue) and CHONCl- (light-orange). The size of each bubble reflects the proportional intensity of the signal from the mass spectra. The embedded doughnut charts quantify class abundance (inner ring, the total number of molecular formulae assigned by mass; outer ring, relative proportions of the chemical families).

conditions under which microbial degradation is suppressed and thermochemical alteration overwhelmingly prevails. Unlike sedimentary diagenesis, these systems lack diagnostic biodegradation markers, and instead show UCM signatures and aromatic/heterocyclic enrichment characteristic of hydrothermal transformation along with the chalcopyrite, further supporting a non-diagenetic mechanism. Although microbial mechanism cannot be entirely excluded, the observed continuum is more plausibly attributed to rapid transformation in hydrothermal settings rather than slow geological diagenesis[51–54].

### Vent shut-down drives bulk nitric molecular enrichment

Molecular formulae obtained by soft negative electrospray ionization (–ESI) Fourier-transform ion-cyclotron-resonance mass spectrometry are derived from an ultrahigh-resolution, non-fragmented technique for hydrothermal organic matter[55–57] that preserves the intact molecular masses of non-biomarker organic compounds consisting of C, H, O, N and Cl (Fig. 4). All extracts show full mass patterns over a range up to *m/z* ~ 1,200. There is a continuum of organic molecules, with a range of carbon oxidation states from nonpolar or slightly polar (polycyclic aromatic hydrocarbons (PAHs), alkylated PAHs and branched/cyclicized/unsaturated molecules) to polar molecules with functional groups (CHO and CHON). The diversity and distribution pattern of organic molecules in different vent extracts is roughly similar but with variation in O/C, H/C, *m/z* and elemental compositions, consistent with that they have been subjected to similarly hydrothermal alterations in Indian Ridge, and their possible genetic linkage. Comparison shows that there is a marked change in both the total number of identified formulae and the relative proportion of N-bearing formulae (Fig. 4, light blue and light orange segments in the doughnut chart), between active and inactive vents at Longqi, Edmond, and Kairei

region, with changing contributions of the various chemical homologous series in elemental compositions (CHO, CHOCl-, CHON and CHONCl-).

At Longqi, active vents with high-temperature hydrothermal activity (351/358 °C) exhibit relatively lower O/C ratios (lower oxidation state) and higher H/C ratios (lipid-trending), indicating enhanced hydrothermal reduction processes. However, the relative proportion of N-bearing formulae remains low (29.9% of >2000 formulae). In contrast, nearby inactive vents show increased O/C ratios, decreased H/C ratios (indicative of condensation), and a significant rise in N-bearing formula proportions (47.9% of >4000 formulae). This suggests that the transformation and enrichment of N-bearing compounds are promoted during the cooling phase as vents become inactive. At Edmond and Kairei, inactive vents show similarly elevated N-bearing formula proportions, with minimal inter-sample variability, yet both slightly exceed Longqi levels. This indicates that regional temperature and activity decreases may drive further N-bearing formula enrichment, enabling stable retention of N-bearing organic structures in vents lacking significant high-temperature (>~300 °C) activity.

A 29.9% → 47.9% → 51.3% rise in the N-bearing formula proportion implies that an inactivating process with decreasing temperature gradient is a key control of hydrothermal nitric molecular enrichment at specific hydrothermal vent sites. After inactivating and cooling, vents favor secondary amination/ammoxidation, adsorption of ammonium, and incorporation of nitrate from seawater, allowing early stabilization of N-rich compounds (e.g., amides; Fig. 3). The shift of CHON points towards elevated O/C and the higher number of identified formulae in inactive vents indicate that N-gain is commonly accompanied by oxygenation, consistent with nitration, hydroxylation, or carboxylation reactions (NSO-enrichment from late Stage 2 to Stage 3; Fig. 3), which also excludes the reverse control of decomposition of CHO on the relative increase in CHON.

In sum, active vents with high temperature are relatively N-bearing structure poor but rapidly enrich N-bearing structures after vent inactivating, whereas ceased vents or vents with no certainly high temperature retain a consistently high N-signature independent of current activity, highlighting specific nitrogen cycling pathways. These findings are essentially consistent with evolutionary relationships shown in the molecular geochemical phylogenetic tree from hard ionization (fragmented) mass spectrometry presented above (Fig. 3).

## Transition from abiotic reduced carbons to prebiotic chemistry

This evolution pathway from chained/branched alkanes through aromatics to heteroatom-rich species, may indicate advanced functionalization and polarity driven by hydrothermal processes. This progression is not random, rather, it suggests a stepwise increase in molecular complexity, providing a basic support for the emergence of life's essential organic feedstock to form building blocks in the primordial Earth.

Ultramafic-hosted hydrothermal systems in Indian Ridge[22] are particularly potent environments for this transformation. They inject $H_2$-rich fluids that promote Fischer–Tropsch-type reactions, generating foundational $CH_4$, short alkanes, PAHs[2,6,39] and even amino acids and thiols (e.g., cysteine)[4,8,19,58,59]. This leads to the formation of amphiphilic molecules bearing reactive C=O and C=N functionalities, key molecular adhesives, directly involved in the formation of amide (peptide) bond and the construction of amino acids, and potentially even nucleobases[60]. The increasing polarity and amphiphilicity of these molecules may also drive emergence of prebiotic substrates and offering catalytic surfaces.

The dynamic thermal and chemical cycling within vents, coupled with mineral catalysis, therefore facilitates not only abiotic carbon reduction but also progressive molecular functionalization. This can result in the metastable, reversible cleavage and reassemblage of linkages like amides, favoring the stepwise assembly and stabilization of peptides and other crucial building blocks[4,8,58–61]. This work provides molecular-level constraints for how hydrothermal systems can reconcile the synthesis, concentration, and hierarchical organization of life's fundamental components within a single, dynamic geodynamic setting, helping to narrow the gap between mantle-drived reduced carbon and biopolymer-ready organics[1,62]. It may represent key actors in the "lipid world" hypothesis[63], and lay the groundwork for the subsequent emergence of the "RNA world[64]" and "protein world[65]".

## Broader implications

These results provide important insights into the preservation and transformation of organic compounds under extreme submarine conditions, and they also establish a robust geochemical foundation for understanding how hydrothermal processes shape organic complexity, highlight potential analogues for prebiotic chemistry in early Earth hydrothermal systems, enabling imagination of origin-of-life scenarios. The shared molecular signatures may also offers a useful framework for identifying molecular biosignatures of deceased or viable life in astrobiological contexts, e.g., the hydrothermal systems on early Mars and other ocean worlds such as Enceladus, Callisto and Europa.

# Methods

The samples' detailed information and mercury data are provided in Supplementary Tables 1 and 2, and more source data are provided as supplementary dataset.

## Sampling description

Hydrothermal chimney samples were collected during the Indian Ocean Hydrothermal Scientific Expedition (TS10 cruise), conducted from November 10, 2018 to March 10, 2019 aboard the research vessel Tansuo-1 (Exploration-1). This expedition lasted 121 days, covered more than 17,000 nautical miles, and deployed the human-occupied vehicle (HOV) Shenhai Yongshi (Deep-Sea Warrior) for 62 dives. This was the first systematic Chinese manned submersible survey that covered both the Southwest Indian Ridge and the Central Indian Ridge[66]. All samples were collected and exported under permits issued by the China Ocean Mineral Resources R&D Association (COMRA) and International Seabed Authority (ISA), in compliance with national and international regulations governing deep-sea scientific sampling. No protected sites or paleontological specimens were involved in this study. The chimney fragment samples described in this study are accessible at the related universities and institutes. Full sample handling chain is documented and the relevant data are deposited in a public repository (see Data availability).

This study focused on samples recovered from three hydrothermal vent fields (Longqi, Kairei, and Edmond) at water depths ranging from ~2700 m to ~3200 m. Among these, the Longqi vent field (also known as Dragon Field) was selected as the primary study site. Additional samples from Kairei and Edmond served as comparative or contrasting references to assess molecular geochemical convergence across geologically distinct systems. Sampling targeted both active and inactive chimneys to ensure cross-site comparability and to evaluate the persistence of molecular and geochemical features under contrasting hydrothermal conditions.

The manipulator arms of Shenhai Yongshi, guided by high-definition seafloor imaging and real-time video monitoring, were used to carefully break off chimney fragments from both walls and conduits. Special attention was given to chalcopyrite-rich inner conduit material, which typically precipitates at >300 °C and thus records high-temperature hydrothermal reactions. To minimize contamination, samples were immediately transferred into titanium containers mounted on the submersible, and were maintained in sealed conditions until shipboard recovery. Upon recovery, samples were then stored at −20 °C prior to molecular and geochemical analyses.

This sampling strategy, covering multiple vent fields, both active and inactive chimneys, and chalcopyrite-bearing inner conduits, ensured representative coverage of Indian Ocean hydrothermal systems, while also providing cross-validation for the geochemical trends observed in the Longqi vent field. Additional samples from Edmond and Kairei were collected to serve as contrasts, allowing assessment of organic geochemical commonalities and differences across vent systems.

Due to field constraints and inherent variability in vent activity, the number of samples per site for this study was necessarily unequal, but the strategy ensures broad coverage of vent types, morphologies, and geochemical regimes and tests natural trends rather than spatial statistical differences.

## Hg concentration and isotope analysis

Total Hg concentration of rock powder samples were determined by an Agilent 7900a, with a detection limit of 0.01 µg/g. Standard reference materials were measured, yielding Hg recoveries of 95–105% and RSD of <7%.

The samples were prepared for Hg isotope analysis using a double-stage tube furnace coupled with 40% anti aqua regia ($HNO_3$/ HCl = 2/1, v/v) trapping solutions. Standard reference materials and method blanks were processed in the same way as the samples. The former yielded Hg recoveries of 95–105% and the latter showed Hg concentrations lower than the detection limit, precluding lab contamination. The preconcentrated solutions were diluted to 0.5 ng/mL with an acid concentration of 10–20% prior to Hg isotope analysis using multi-collector inductively coupled plasma mass spectrometry (MC-ICPMS, Neptune Plus, Thermo Scientific, and Nu plasma 3D, Nu instruments Ltd.). For each sample, multiple measurements were performed to ensure signal stability and to calculate a robust average and standard deviation. The reported isotopic values represent the average of these replicate measurements.

MDF is expressed in $\delta^{202}Hg$ notation in units of‰ referenced to the NIST-3133 (analyzed before and after each sample). Hg isotope ratios were reported following the conventionusing $\delta$ notation defined by the following equation:

$$\delta^{xxx}Hg(\%) = [(^{xxx}Hg/^{198}Hg)_{sample}/(^{xxx}Hg/^{198}Hg)_{standard} - 1] \times 1000 \quad (1)$$

where $^{xxx}Hg$ is $^{199}Hg$, $^{200}Hg$, $^{201}Hg$, $^{202}Hg$, MIF is reported in $\Delta$ notation, which describes the difference between the measured $\delta^{xxx}Hg$ and the theoretically predicted $\delta^{xxx}Hg$ value, in units of ‰:

$$\Delta^{xxx}Hg = \delta^{xxx}Hg - \delta^{202}Hg \times \beta \quad (2)$$

$\beta$ is 0.252 for $^{199}Hg$, 0.5024 for $^{200}Hg$, and 0.752 for $^{201}Hg$. Hg concentration and acid matrices in the bracketing NIST-3133 solutions were matched with neighboring samples. NIST-8610 secondary standard solutions, diluted to 0.5 ng/ mL Hg with 10% HCl, were measured every 7 samples. Standard reference material NIST-3133 was prepared and measured in the same way as the samples. Analytical uncertainty was estimated based on replicate analyses of the NIST-3133 secondary standard solution and full procedural analyses of NIST-8610.

## Total organic carbon
The TOC was measured using an Elementar Vario EL cube EA with decarbonated samples (5–10 mg). Quality control checks were performed, including sample duplicates, analysis of certified reference materials (acetanilide), and analysis of blanks. The measurement precision was better than ±0.1 wt % for TOC.

## Solvent extraction
Organic-lean samples are highly susceptible to contamination from external sources during both field collection and laboratory processing. To ensure the indigeneity of hydrocarbons extracted from chimneys, all preparation steps were carefully monitored for potential contamination, including solvents, glassware, evaporation systems, cutting tools, sample handling materials, etc[67–69]. All procedural blanks showed no sterol-related ions above background noise (Supplementary Fig. 9a), demonstrating that sterols observed in samples are not introduced during workup.

To preserve native yet mostly shared molecular features in sulfide-rich, very low-TOC chimney samples, we used high-purity solvent extraction only. After method trials, a 1:1 DCM:MeOH mixture was selected as it maximized molecular diversity (non-polar to polar NSO). All the samples that were treated were held using combusted aluminum foil in order to minimize the introduction of surficial contamination from the laboratory. The bulk rocks were washed for few seconds by sonication in deionised water and dichloromethane to remove surface contamination and to avoid laboratory and cross contamination, then were dried in fume cupboard at room temperature, and were cut outside surfaces, and the saw was rinsed with Millipore water and DCM between each sample to avoid cross contamination, and the most inner parts (10-30 g) were crushed into powder in a rotary mill, and the mill was cleaned between crushing each sample by scrubbing under water and rinsing with DCM 10 times. The homogenized powdered and hard solid parts were loaded and extracted using a Soxhlet apparatus with a mixture of dichloromethane-methanol (1:1 v/v) for 72 h, to enhance the extractability.

## Gas chromatography-mass spectrometry
Gas chromatography-mass spectrometry (GC-MS) analyses of the extractions were performed on an Agilent GC (6890) coupled to an Agilent Mass Selective Detector (5973) equipped with a J&W HP-5MS (5% phenyl methyl siloxane) fused silica column (30 m × 0.25 mm × 0.25 µm). The inlet temperature was 280 °C. Samples (1 µL) were injected in splitless mode. The temperature of the GC oven was initially held at 40 °C for 1 min., programmed to 290 °C at 4 °C/min., and held for 24 min. Helium (99.999%) was used as the carrier gas, with a constant flow rate of 1.0 mL/min. The ion source of the MS was operated in EI mode at 70 eV. The MS data were acquired in full scan mode. Standards (AccuStandard, Inc, only for PAH parallel check), procedural blanks, solvent blanks and burning blanks were performed repeatedly until the signal was stable and make sure there is a clean background with blank shots. Contamination was evaluated by monitoring diagnostic ions in airborne, burning, and procedural blanks (Supplementary Fig. 7). As shown in Supplementary Fig. 9a, the blanks showed $\times 10^2$ level signals at the characteristic retention times, confirming that the discussed mass spectra detected in chimney extracts are indigenous to the hydrothermal samples rather than introduced during workup. This ensured the extraction and instrumental procedures were reliable and repeatable for non-target compound detection.

## Fourier transform mass spectrometry
Preliminary attempts with atmospheric pressure photoionization were also tested, but due to the low TOC contents (< 0.1 mg/g), the extract concentrations did not reach the typical requirement (~0.5 mg/mL), and thus no effective signals were detected. Therefore, only ESI spectra are performed. The mother liquor concentration was 10 mg/mL in toluene, and injection was performed after dilution to 0.2 mg/mL using a toluene:methanol (1:3) mixture. The extracts were analysed using an Orbitrap Fusion MS (Thermo Scientific, USA) mass spectrometry (–ESI FT-MS). Samples were infused directly into the negative mode ESI sources at a speed of 180 µL/h. The ranges were $m/z$ 200 – 1500. The instrument resolution mode was selected as 500,000. Ion spray voltage 2600, sheath gas 5, aux gas 2, ion transfer tube temperature 300 °C, RF-lens 60%, modified AGC target 1,000,000, Maximum Injection Time 100 ms were acquired. To ensure data reliability, replicate analyses and quality controls were conducted. A standard reference sample (Suwannee River Fulvic Acid, SRFA) was analyzed three times by the same operator in one session, yielding repeatability RSD values of relative peak intensities <5%. Reproducibility was further evaluated across 26 replicate measurements performed by different operators over six months, with RSDs of average molecular parameters generally <3%. These results confirm that replicate analyses produce stable and reproducible molecular information.

## Mass spectra, identification, co-annotation, similarity analysis
The raw Agilent GC−MS data files that were obtained from the GC−MS system were initially converted to an open, XML-based format suitable for mass spectrometer output files from different instrument vendors, which was configured to fit the computing platform and to facilitate rapid and efficient data retrieval for subsequent analysis. The mass spectral data were then subjected to deconvolution using an open-customized program with a built-in deconvoluting feature, which was specifically designed for assessing untargeted organic molecules, and is capable of handling data-independent acquisition data.

All co-annotation processing and molecular similarity analyses were conducted on a workstation equipped with an Intel Xeon Sapphire Rapids CPU (12 cores, 24 threads, base clock 2.59 GHz, max TDP 270 W, 48 MB L1 + 12 MB L2 + 30 MB L3 cache) and an NVIDIA RTX A6000 GPU (GA102 architecture, 48 GB GDDR6 memory, 10752 CUDA cores, 768.0 GB/s memory bandwidth). The GPU supported advanced acceleration features including CUDA, DirectML, Vulkan, and ray tracing technologies, which were utilized for deep similarity computation and matrix optimization. The system ran on Windows 11 with NVIDIA driver version 32.0.15.7216 (WHQL certified). This configuration ensured efficient and effective (the simultaneous processing of 12 samples without memory overflow or computational bottlenecks)

handling of high-dimensional molecular data, large-scale annotation tasks, large-scale mass spectral deconvolution, co-annotation, and molecular similarity network construction, with stable precision and rapid convergence. The computational load of simultaneous co-annotation and similarity analysis across 12 samples provides sufficient parallel processing capacity and memory bandwidth for stable data throughput. Hard ionization type, centroid data type and positive ion mode were selected for the process. The programmed data collection ranges were $m/z$ 0–1000 Da. The program performed peak detection (spotting) by exploring two data axes of retention time and accurate mass ($m/z$), analyzed the mass spectra, and identified significant peaks above the baseline noise (or customized minimum peak height). The noise level threshold (or customized minimum peak height) was set to a value of around $3 - 4 \times 10^5$ that was appropriate for the specific data sets. Linear-weighted moving average was used for the peak detection by default to accurately determine the peak left- and right-edges, and the smoothing method was set to level 3.0. The mass deconvolution within the program was applied to each detected ion spot for deconvolution (Supplementary Fig. 2). This process involved the extraction of product mass spectra and the reconstruction of pure mass spectra, which was based on the ideal slope and sharpness values of the model peaks. The sigma window value was set to 0.5, and the EI spectra cut off was set to 10.0.

The representative deconvoluted (PURIFIED) EI spectrum was automatically selected as the spectrum of the highest identification score for all samples aligned to the focused alignment spot. We compared and matched the representative deconvoluted EI spectra with reference EI spectra from mass spectral libraries, e.g., NIST20, MassBank, LipidBlast and other publicly available databases. Compound identification was based on a similarity scoring system that considered mass spectral similarity, accurate mass, and retention time.

To ensure the statistical robustness and reproducibility of our untargeted analysis, we performed a cross-validation procedure on the data processing. The entire dataset was processed and analyzed three separate times, using progressively stringent total spectral similarity score thresholds of 20%, 50%, and 70% for compound co-annotation. This triple-check process allowed us to confirm that the key molecular clusters and the overall topology of the resulting geochemical phylogenetic tree were stable and not artifacts of a specific parameter choice. For the final analysis presented in the manuscript, a minimum spectrum similarity score of 70% and an m/z tolerance of 0.5 Da were set as the final thresholds for organic compound identification. After deconvolution of the mass spectra, data were accurate enough to be reliably compared across different samples. The program executed peak alignment for different samples in different locations. The alignment process was based on retention time, with an additional EI similarity tolerance. The retention time tolerance for alignment was set to 0.075 min and the total spectrum similarity score was set to > 70%, as in some cases the use of a wide retention time tolerance (e.g., 0.5–1 min) could lead to wrong peak alignment of isomers. The retention time factor and EI similarity factor were both set to 0.5. This step was to analyse the comparability and chemical association of organic compounds commonly present in different sample locations, which supported the subsequent molecular phylogenetic analysis. Raw metadata similarity matrix or spectral information were then exported for phylogenetic analysis.

### The phylogenetic analysis and visualization
A hierarchical clustering tree, presented as a molecular-fingerprint-based maximum likelihood phylogenetic tree of organic compounds was calculated using the alignment and similarity metadata of 654 organic molecules (233 identified with mass spectral similarity > 70% to NIST20 library; and the rest are unknown compounds). The phylogenetic tree were visualized and refined with open visualization program. This tree represents a statistical clustering of molecules based on molecular structural similarity inferred from their mass spectral fragmentation patterns, analogous to phylogenetic methods used in metabolomics, but reflecting geochemical relationships rather than biological ancestry.

Detailed analysis scripts are provided in Supplementary codes and data.

### Reporting summary
Further information on research design is available in the Nature Portfolio Reporting Summary linked to this article.

## Data availability
The source data generated in this study are provided with this paper in the Supplementary information/source data file. The raw data generated in this study have been deposited in the Zenodo database[70] available at https://doi.org/10.5281/zenodo.17398264. Source data are provided with this paper.

## Code availability
All codes are presented in the Supplementary information.

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

## Acknowledgements

This research was supported by the National Natural Science Foundation of China (Grant Nos. 42488101 [Q.L.] & 42422205 [H.X.]). Q.L. acknowledges the support from the Tencent Foundation of China through the XPLORER PRIZE.

## Author contributions

Q.L., H.X., J.W. and Z.J. conceived the study. Q.L., H.X., Z.J., C.Z., S.L. and Z.J. performed the experimental studies and carried out the analysis. Q.L., H.X., J.W., J.L., D.Z., C.T., S.L., H.N., F.H. and Z.J. performed the research. H.X., C.Z., Y.F., B.Z., G.Y. and S.X. performed data curation and visualization. Q.L., H.X., D.Z., F.H. and Z.J. supervised the work and project. Q.L., H.X., F.H. and Z.J. acquired funding. All authors contribute to writing, original draft, review and editing.

## Competing interests

The authors declare no competing interests.
