## [Transparent Peer Review file · Nature Communications]

Abyssal hydrothermal alteration drives the evolution from simple alkanes to prebiotic molecular complexity

Corresponding Author: Professor Quanyou Liu

Version 0:

Reviewer comments:

Reviewer #1

(Remarks to the Author)

Abyssal hydrothermal alteration drives the evolution from simple alkanes to prebiotic molecular complexity

Overview and general recommendation

Dear Editor and dearest authors,

I first want to acknowledge the opportunity to read and give my comments on the article "Abyssal hydrothermal alteration drives the evolution from simple alkanes to prebiotic molecular complexity" authored by Liu et al.

In such article, authors proposed a mechanism to explain the possible way to reach a high molecular complexity from simple molecules, before the emergence of life. The approach is very innovative and relies on the idea that there is a natural progression on the composition and structure of molecules in key environments that originated the simple elements that triggered the origin of life. Authors suggest that this path can be revealed through a systematic analysis.

As a model, authors explored the diversity of organics in an abyssal hydrothermal system. In particular, they analysed the Edmond, Kairei, and Longqi vent fields, located in Southwest Indian Ridge (SWIR). This system is considered an ultraslow spreading system. The main idea is to outline a hypothesis about how the multiple variables in these environments drive the accumulation and evolution of organics, from alkanes to complex organic compounds and, ultimately, to the building blocks of biomolecules. I have some comments and questions that I hope will help improve the manuscript before it is accepted.

A. Originality and significance: if not novel, please include reference

This article presents a new approach in the study of the molecular complexity required prior to the emergence of life. The main idea is based on a parallelism of metabolomic and taxonomic analyses applied to organics. This is a new way to study the organics in the prebiotic chemistry area, including a bulk analysis, and tracing a path to elucidate how organics could have evolved. Data presented are novel and themselves constitute a good contribution to scientific knowledge.

B. Data & methodology: validity of approach, quality of data, quality of presentation

I have some questions in this regard. I search the sampling strategy and the representativity of samples, and I couldn't find it. I think it is very relevant to describe at most detail this part, as it is fundamental to follow the main idea of the article. For example, I wonder if sampling was enough and if the data do not present a bias. Could you be so kind to include the sampling method and a description of sampling?

In addition, please, explain why do you have a such different number of samples in each collecting site? There are 12 points, and 7 are from active points and 5 from inactive ones. The number of samples in each location is not the equitable. Do you consider these differences could influence your results?

In general, figures are clear and fine. The quality of presentation of data is fine. I just recommend checking the Figure 4, the bars (at the bottom) look distorted.

C. Appropriate use of statistics and treatment of uncertainties

There is not a description of the employed statistics. Please, describe the number of measurements made, if there are replicas, and the statistical methods.

D. Conclusions: robustness, validity, reliability

There are not conclusions in the manuscript.

E. Suggested improvements: experiments, data for possible revision

According to the information provided, just the solvent extractable fraction was analysed. Is there any reason for this? Have you ever considered that another release method (the use of strong bases) might be useful? If possible, please essay this treatment, it possibly could give more data.

Even if the protocol for the extraction of organics is described in the methods section, the amount of each sample treated is not explicit nor the volume of solvents employed. In addition, it is not said if replicas were made for each analysis (Hg isotope, CG-MS, FT-ICR MS). Please clarify the previous point.

The procedure for the FT-ICR MS is well described. However, I missed a detailed description of the samples analysed by FT-ICR MS. How much extract was analysed? Did you estimate before the concentration of organics in samples? Why did you select ESI and not API for study your samples? I understand that ESI is better for the analysis of polar and high molecular weight molecules. Nonetheless, many biomarkers are lipids and consequently, some of them are non-polar or slightly polar, so probably you could find nice information by using API. I guess if you performed the two analysis and chose the ESI instead of API for getting better results? If that is the case, please include the information. If not, it could be interesting to include those analyses in your samples.

F. References: appropriate credit to previous work?

The references are appropriated, and the article is well documented, there is a bunch of articles in the HV area and authors selected key documents. I would just suggest including some references of the application of FT-ICR MS in other studies to trace the reach and possible meaning. For example:

- Butturini, A., Amalfitano, S., Herzsprung, P., Lechtenfeld, O. J., Venturi, S., Olaka, L. A., ... & Fazi, S. (2020). Dissolved organic matter in continental hydro-geothermal systems: insights from two hot springs of the East African Rift Valley. *Water*, 12(12), 3512.
- Ventura, G. T., Rossel, P. E., Simoneit, B. R., & Dittmar, T. (2020). Fourier transform ion cyclotron resonance mass spectrometric analysis of NSO-compounds generated in hydrothermally altered sediments from the Escanaba Trough, northeastern Pacific Ocean. *Organic Geochemistry*, 149, 104085.
- Gomez-Saez, G. V., Niggemann, J., Dittmar, T., Pohlabein, A. M., Lang, S. Q., Noowong, A., ... & Bühring, S. I. (2016). Molecular evidence for abiotic sulfurization of dissolved organic matter in marine shallow hydrothermal systems. *Geochimica et Cosmochimica Acta*, 190, 35-52.

G. Clarity and context: lucidity of abstract/summary, appropriateness of abstract, introduction and conclusions

The manuscript is well written and there is a subdivision considering the main topics. The abstract is concise and gives information of the key findings of the research. However, there is not an introduction nor conclusion sections. Authors present a section called Broader significance at the end of the Discussion. However, I consider that they could include a final section (Conclusion) comprising the relevance of their data, the implications for prebiotic chemistry, their future work, etc.

Further Questions

I consider very appealing the idea of tracing the evolution of molecules in HV; however, it must be considered with caution. Taxonomy (hierarchical classification) is not the same that Phylogenetics (study of the evolutionary history) or Genomics (the study of structure, function, evolution of the genomes). You stay that "For example, genomics supplies a static hereditary framework, yet hydrothermal alteration and transformation of organic compounds present a dynamic scaffold of geochemical information". In fact, the genomes are constantly evolving, and evolutionary hypotheses (phylogenetic trees) establish the historical relationship between organisms. Biologists never consider genetic information to be static.

You mentioned that it is important to understand the mechanisms that allowed the formation and self-assembly of organic molecules under changeable conditions, including chemical and physical variables (i.e. pH, temperature, chemical composition, redox state, etc.). This is one of the most important questions in the study of hydrothermal vents as niches for prebiotic evolution. However, I do not find in your text a reflection about this, or the incorporation of some of those variables in the analysis of your data. In this sense, it could be important to have all the data of the sampling sites (Supp. Table 1); please, complete the information.

Hydrothermal processes—including temperature, pressure, chemical gradients, Eh,pH, etc.— are responsible for greatly altering the organic molecules found in these environments. In your experience, how could we differentiate between organic

molecules formed from simple precursors in HV and those resulting from the decomposition of complex organic matter from organisms? The remains of different organisms could be reaching the ocean floor through different means, even though these organisms do not live near HV environments. The question relies on the fact that, if we could distinguish between those processes we could reinforce your hypothesis of molecular evolution.

Hydrothermal vents during the Hadean Eon must have been shallower than contemporary submarine vents. Could you explain how we can determine the similarity between these systems?

Figure 2 represents the distribution of organic compounds in a representative sample (SY139-G07; Edmond). Please, include another sample from an active site in the same figure. It could be also interesting to have the other data in Supplementary Content.

Minor comments

Regarding the data obtained from FT-ICR MS, please include the list of all identified molecules in the supplementary material (probably in an Excel spreadsheet).

The supplementary material must include all the data presented in Figures, both in the main text or in the supplementary information; please incorporate this information.

Please check the text and correct the word earth (as a planet), replace it with Earth.

Line 378. Please give a temperature range in the sentence "significant high-temperature activity".

Line 409. Remove the point after "(e.g., cysteine)4,8,19,51,52., and".

Line 452- Correct the subindex in (HNO₃).

Reviewer #2

(Remarks to the Author)

The abstract and paper needs major rewrite for more general science audience.

Rewritten abstract as follows

Abyssal hydrothermal vents are regarded as crucibles for prebiotic organic chemistry. While genomics and metabolomics have been extensively studied in such settings, the organic geochemical continuum and evolutionary transitions that link simple abiotic reduced carbon to heteroatom-rich polymer precursors remain poorly constrained, largely due to the highly dynamic hydrothermal alteration and transformation.

Here, we apply a metabolomics-inspired strategy incorporating both hard and soft ionization, algorithmic deconvolution of mass spectra, high-precision structural identification and molecular similarity matching, and hierarchical organization, to construct a molecular-relatedness organic geochemical phylogenetic tree of vents from ultraslow-spreading Southwest Indian Ridge. We report the shared evolutionary relationship among organic molecules across different vent sites of differing geologic activity.

Molecular progressive reconstruction of alkanes, alkanols, phenyls, PAHs, NSO-compounds, suggests that, regardless of abiotic or biotic origin, these molecules have been extensively transformed under hydrothermal conditions. The observed molecular evolution from simple alkanes to complex heteroatom bearing compounds particularly nitrogenous species accumulating during vent shutdown- reveals a "heteroatomization" trajectory that signals increasing functionalization and polarity in the Hadean ocean. This finding helps bridge the gap between abiotic reduced carbon compounds and life's essential organic feedstock, further underscoring the role of hydrothermal systems in mediating the transition from abiotic molecules to prebiotic chemistry. It provides a foundational pathway for the emergence of life's building blocks on the primordial Earth

I found this paper very difficult to follow- the overall aims needs carefully articulating.

This study investigates the molecular evolution of organic compounds in abyssal hydrothermal vents, environments thought to be key sites for prebiotic chemistry. Despite well-documented genomic and metabolomic studies in such settings, the transition from simple abiotic molecules to complex prebiotic organics remains poorly understood due to intense hydrothermal alteration. Using a metabolomics-inspired strategy—including advanced mass spectrometry, algorithmic spectral deconvolution, and molecular similarity mapping—the authors constructed a geochemical phylogenetic tree from vents along the ultraslow-spreading Southwest Indian Ridge. They identified evolutionary links among diverse organic molecules (e.g., alkanes, alcohols, polycyclic aromatics, and nitrogen/sulfur/oxygen-containing compounds), revealing a continuum from simple hydrocarbons to more complex, heteroatom-rich structures. These transformations occur irrespective of the molecules' biotic or abiotic origins, with shutdown phases of venting showing an increased presence of nitrogenous compounds. This progression reflects a molecular "heteroatomization" pathway, suggesting increasing functionality and polarity—key traits for life's chemical precursors. The findings highlight hydrothermal systems as crucibles for the stepwise evolution of reduced carbon toward biologically relevant molecules, helping to bridge the gap between abiotic chemistry and the emergence of prebiotic organics on early Earth.

It is very difficult to assess the quality of data generated from the samples investigated. The input data needs to be vigorously validated - from the origin of the samples, preparation of samples, procedural blanks through the whole process needs careful consideration. The samples cover a range of compounds -polarity and some of these compounds are likely introduced contaminants from storage-to analyses. This is my biggest concern with the data. More background information is needed before this can be published.

Version 1:

Reviewer comments:

Reviewer #1

(Remarks to the Author)

Dear authors and Editor,

I hope that you are doing well. I am sending to you my second review to the manuscript titled "Abyssal hydrothermal alteration drives the evolution from simple alkanes to prebiotic molecular complexity", authored by Liu and coauthors. The article proposes a hypothesis about the progression in the composition and structure of organic molecules, driven by likely natural processes in primitive environments. As an example, they propose the study of organics in HVS. The approach is novel and could certainly be a good topic for discussion in the scientific community.

In my previous comments, I raised some points that could improve the manuscript. The authors re-elaborated their manuscript and addressed all my earlier suggestions thoroughly. I greatly appreciate the time taken to respond and the in-depth answers.

Also, the required data and graphic material were properly provided, analysed and presented.

The authors explained the methods at detail, as well as the reasons that underpin their decisions regarding sampling, analytical techniques, use of statistics, and data analysis. The result is a clearer and more robust manuscript. I very much acknowledge the inclusion of the photographs of samples and raw data; this is great for readers.

I have no more comments to the manuscript, just say that it represents a good contribution to the academic discussion.

Reviewer #2

(Remarks to the Author)

Unfortunately the manuscript has not satisfied concerns with regards to contamination. None of the procedural blanks and analyses have been included in the manuscript. This causes concerns on the origin of compounds and even more so when samples have extremely low TOC%.

Version 2:

Reviewer comments:

Reviewer #2

(Remarks to the Author)

I have reviewed the authors' response regarding contamination and I am largely satisfied with the revisions. However, I still have some concerns that the sterols may have been introduced during the workup procedures. Sterols are common laboratory contaminants and can easily be absorbed onto samples. I would like the authors to be fully certain that the sterols reported here are genuinely indigenous.

A continuum of related compounds has previously been observed in a Devonian crustacean fossil, where the entire continuum was attributed to microbial transformations. The authors should include a reference to this study (Melendez et al., 2013, Scientific Reports) and briefly discuss in the manuscript other processes that may account for a diagenetic continuum.

We thank two anonymous reviewers and their helpful comments and constructive feedback. The list of changes and our **Point by Point Response** to the two reviewers can be found in the **blue text** below, and **Quotes** from our revised text in the manuscript are in **green**. All suggestions and opinions were fully considered and accepted. Corrections and revisions have been carefully made. Some comments and encouragement that were made will be of great help for our continued work, and are much appreciated. Please note that this is **a much revised manuscript, the line numbers and the figures in the manuscript have greatly shifted** after revision, and the change of citations by **Endnote** cannot be tracked by **office word program**, and the changes in tables, figures and other materials may not be tracked either. In some cases, the answer in the response to similar questions may be slightly **repetitive, overwritten or combined**. We have submitted a copy of the *Revised Manuscript with Track Changes* and a **clean copy of the Manuscript file**. The *Supplementary Information* (a clean copy and one with Track Changes) were uploaded as well.

REVIEWER COMMENTS

List of changes response to Reviewer #1

Reviewer #1 (Remarks to the Author):

Abyssal hydrothermal alteration drives the evolution from simple alkanes to prebiotic molecular complexity

Overview and general recommendation

Dear Editor and dearest authors,

I first want to acknowledge the opportunity to read and give my comments on the article “Abyssal hydrothermal alteration drives the evolution from simple alkanes to prebiotic molecular complexity” authored by Liu et al.

In such article, authors proposed a mechanism to explain the possible way to reach a high molecular complexity from simple molecules, before the emergence of life. The approach is very innovative and relies on the idea that there is a natural progression on the composition and structure of molecules in key environments that originated the simple elements that triggered the origin of life. Authors suggest that this path can be revealed through a systematic analysis.

As a model, authors explored the diversity of organics in an abyssal hydrothermal system. In particular, they analysed the Edmond, Kairei, and Longqi vent fields, located in Southwest Indian Ridge (SWIR). This system is considered an ultraslow spreading system. The main idea is to outline a hypothesis about how the multiple

variables in these environments drive the accumulation and evolution of organics, from alkanes to complex organic compounds and, ultimately, to the building blocks of biomolecules. I have some comments and questions that I hope will help improve the manuscript before it is accepted.

A. Originality and significance: if not novel, please include reference

This article presents a new approach in the study of the molecular complexity required prior to the emergence of life. The main idea is based on a parallelism of metabolomic and taxonomic analyses applied to organics. This is a new way to study the organics in the prebiotic chemistry area, including a bulk analysis, and tracing a path to elucidate how organics could have evolved. Data presented are novel and themselves constitute a good contribution to scientific knowledge.

Thanks for the comments.

B. Data & methodology: validity of approach, quality of data, quality of presentation

I have some questions in this regard. I search the sampling strategy and the representativity of samples, and I couldn't find it. I think it is very relevant to describe at most detail this part, as it is fundamental to follow the main idea of the article. For example, I wonder if sampling was enough and if the data do not present a bias. Could you be so kind to include the sampling method and a description of sampling? In addition, please, explain why do you have a such different number of samples in each collecting site? There are 12 points, and 7 are from active points and 5 from inactive ones. The number of samples in each location is not the equitable. Do you consider these differences could influence your results?

■ We sincerely thank the reviewer for this important comment, which allows us to clarify our sampling strategy. We acknowledge and we agree that a more detailed description is necessary and have now added a detailed description of sample collection in the Methods section (see Methods - Sampling description section).

“Sampling description

Hydrothermal chimney samples were collected during the Indian Ocean Hydrothermal Scientific Expedition (TS10 cruise), conducted from November 10, 2018 to March 10, 2019 aboard the research vessel Tansuo-1 (Exploration-1). This expedition lasted 121 days, ...

...”

Our sampling strategy was designed to maximize representativity while considering the geological constraints of hydrothermal vent systems. Sampling was conducted at

three vent fields along the Southwest Indian Ridge (SWIR): Longqi, Edmond, and Kairei. The selection of sites was based on both hydrothermal activity and geological accessibility, aiming to test whether molecular transformation patterns observed at Longqi are reproducible across diverse vent settings. The spatial coverage and structural diversity of sampling sites were designed to reflect hydrothermal system variability rather than achieve numerical symmetry.

All samples were directly recovered from such mineralized chimney structures, rather than surrounding sediments or low-temperature discharge zones. Samples were collected from the inner conduits of black smoker chimneys, particularly chalcopyrite-rich zones that precipitate from high-temperature vent fluids. These portions are most representative of the hydrothermal reactions occurring within active conduits and thus are crucial for reconstructing molecular alteration.

Ensuring coverage across different vent fields and activity states. To avoid site-specific bias, we collected samples from three hydrothermal fields (Longqi, Kairei, and Edmond), including both active and inactive chimneys. This comparative design allows us to test whether the molecular geochemical features identified at Longqi also occur in other locations, thereby assessing their generality across the Indian Ocean ridge system. Longqi served as our principal case study, while Kairei and Edmond provided contrastive testing of geochemical convergence and molecular complexity evolution across different locations.

We appreciate this number concern. The unequal number of samples reflects the intrinsic spatial heterogeneity of hydrothermal chimney systems, and accessibility during HOV operations, rather than a bias in our sampling design. Chimney differ markedly in activity, size, and preservation state, which naturally limits the number of effective samples that can be collected at each site.

Importantly, our research aims to identify the common molecular geochemical signatures shaped by hydrothermal processes, rather than to compare microbial community differences between regions or vent activity states, or to achieve a statistically uniform sampling scheme or statistical equalization. Thus, the variations in sample numbers do not bias the conclusions. On the contrary, the inclusion of both active and inactive chimneys from multiple vent fields provides robust cross-validation (controls) that the observed molecular evolutionary features are indeed general and not site-specific, and to test the persistence. The phylogenetic tree approach we employed is specifically designed to handle heterogeneous samples and to extract reproducible, system-wide features from complex backgrounds. This means that the conclusions rely on shared evolutionary signals, not on the absolute number of compounds detected at each site. From this perspective, the uneven number of samples across sites does not bias our results but rather reinforces the robustness of the MOST universally shared molecular patterns we report. In essence, many more compounds are present but not co-annotated in our phylogenetic framework, and thus are not included in the phylogenetic tree. Since our goal is to identify commonalities, this inherent heterogeneity makes the dataset more representative and prevents bias

from any single region.

We therefore believe that the combination of representative high-temperature samples with cross-field and cross-activity comparisons provides a robust basis for our conclusions. The revised Methods section now explicitly describes this strategy to ensure clarity for readers.

We hope this clarification addresses the reviewer's concern. We appreciate the opportunity to strengthen this critical aspect of the manuscript.

In general, figures are clear and fine. The quality of presentation of data is fine. I just recommend checking the Figure 4, the bars (at the bottom) look distorted.

Fig. 4 was fixed. It should be the problem of pdf transformation.

I provided a .pptx version for better conversion after uploading.

C. Appropriate use of statistics and treatment of uncertainties

There is not a description of the employed statistics. Please, describe the number of measurements made, if there are replicas, and the statistical methods.

■ Thanks for the question.

We appreciate the reviewer's comment regarding the description of statistical treatment. In our revised manuscript, we have added detailed information in the *Methods* and *Supplementary Information* sections and other relevant parts.

Our study combines two distinct types of geochemical data: (1) exploratory, untargeted organic molecular data e.g., GC-MS, and (2) quantitative, single-value data e.g., Hg isotope analyses. The statistical validation for each data type is necessarily different

For both Hg concentrations and isotopes, replicate measurements of international standard reference materials (NIST-3133, NIST-8610) yielded recoveries and relative standard deviations. Method blanks were consistently below detection limit, ensuring that no contamination was introduced during sample preparation. Replicate analyses of the same sample were carried out until stable values were obtained, and uncertainties are reported as 2SD based on repeated measurements of standards and samples.

See Supp. Table 2 and "*Hg concentration and isotope analysis*" section.

Region	Activity	Sample codes	Hg (ppb)	$\delta^{199}\text{Hg}$ (‰)	2SD	$\delta^{200}\text{Hg}$ (‰)	2SD	$\delta^{201}\text{Hg}$ (‰)	2SD	$\delta^{202}\text{Hg}$ (‰)	2SD	$\Delta^{199}\text{Hg}$ (‰)	2SD	$\Delta^{200}\text{Hg}$ (‰)	2SD	$\Delta^{201}\text{Hg}$ (‰)	2SD
Longqi	Active	SY111-G10	103.2	-0.077	0.070	-0.255	0.038	-0.370	0.056	-0.492	0.071	0.047	0.067	-0.008	0.031	0.000	0.043
	Active	SY097-G01															
	Active	SY105	963.4	-0.136	0.070	-0.388	0.038	-0.561	0.056	-0.786	0.071	0.062	0.067	0.007	0.031	0.030	0.043
	Active	SY104-G06															
	Active	SY109-G07															
Edmond	Inactive	SY093-G07	8.5	-0.349	0.070	-0.751	0.038	-1.108	0.056	-1.550	0.071	0.042	0.067	0.028	0.031	0.057	0.043
	Active	SY107	1841.0	-0.115	0.070	-0.317	0.038	-0.493	0.056	-0.656	0.071	0.051	0.067	0.013	0.031	0.001	0.043
	Inactive	SY139-G7	473.8	-0.201	0.070	-0.391	0.038	-0.573	0.056	-0.788	0.071	-0.002	0.067	0.005	0.031	0.020	0.043
	Inactive	SY134-G06	13390.0	-0.164	0.070	-0.454	0.038	-0.652	0.056	-0.945	0.071	0.074	0.067	0.021	0.031	0.058	0.043
	Inactive	SY150-G6															
Kairui	Active	SY147-G8	316.7	-0.137	0.070	-0.250	0.038	-0.435	0.056	-0.579	0.071	0.009	0.067	0.041	0.031	0.000	0.043
	Inactive	SY145-G1	2460.0	-0.084	0.070	-0.198	0.038	-0.272	0.056	-0.335	0.071	0.000	0.067	-0.029	0.031	-0.020	0.043

For targeted geochemical measurements, e.g., single-value parameters such as Hg isotope ratios or TOC, each sample was analyzed at least in duplicate, with repeated measurements producing stable values within the analytical uncertainty. For TOC analysis, measurement precision was determined through sample duplicates and the analysis of certified reference materials (acetanilide). As stated in the Methods, the reported precision was better than ± 0.1 wt%. See “Total organic carbon” section

For untargeted organic compound detection, e.g., GC-MS and FT-ICR-MS analyses, the purpose is not to precisely quantify each of the thousands of individual peaks, but to do *identification and structural comparison* of molecules to characterize systematic transformation pathways and molecular patterns (i.e., the geochemical phylogenetic tree). Following the established best practices in organic geochemistry, reproducibility was ensured during method establishment by running standards and blanks repeatedly until the signal was stable and baseline is clean, establishing stable background and repeatable signal profiles. This foundational work ensures the reliability of the signals detected in the actual samples. Once stabilized, our prepared samples were analyzed to obtain raw data, which were subsequently processed through peak detection, spectral deconvolution and compound co-annotation. Specifically, the reproducibility and repeatability of FT-ICR MS results were systematically evaluated through replicates (quality control samples were analyzed multiple times within the same session and across different operators and time periods, showing stable results with low relative standard deviations, including both intra-day and inter-day evaluations), ensuring the robustness and consistency of the data across time and operators.

During co-annotation and similarity analysis, to cross-validate reproducibility, spectral similarity thresholds were applied at three levels (20%, 50%, 70%) and only compounds with $>70\%$ similarity were retained for final interpretation. This cross-validation confirmed that the key molecular groupings and the overall topology of our phylogenetic tree were stable and not artifacts of a specific parameter choice. This approach ensures that the patterns we report are statistically sound and reproducible from the metadata. Error propagation is not applicable to such large-scale untargeted profiling, all procedures followed recognized best practices in molecular geochemistry, as thousands of compounds are detected, it is neither practical nor standard to perform triplicate quantification for each peak (mostly convoluted and unknown), instead, reproducibility is addressed through metadata similarity analysis and co-annotation across samples. Statistical matching was cross-checked across all samples. For untargeted molecular data, uncertainties are addressed by applying strict thresholds in spectral similarity and retention-time alignment, and by using multiple

cut-offs for cross-validation. Metadata derived from these co-aligned peaks formed the basis of the phylogenetic molecular fingerprinting. This procedure inherently incorporates internal data reproducibility, as only compounds consistently aligned across samples are retained. This ensures that the reported molecular patterns are robust and reproducible.

See “*Mass spectra, identification, co-annotation, similarity analysis*” section. We also upload the co-annotation reproducibility in Supp. Tables

“To ensure the statistical robustness and reproducibility of our untargeted analysis, we performed a cross-validation procedure on the data processing. The entire dataset was processed and analyzed three separate times, using progressively stringent total spectral similarity score thresholds of 20%, 50%, and 70% for compound co-annotation.....”

Other changes see *Methods* section.

D. Conclusions: robustness, validity, reliability

There are not conclusions in the manuscript.

We appreciate the reviewer’s later suggestion, and have revised the manuscript to include a new **Conclusion** section. To avoid redundancy, we merged the previous *Broader significance* with the new *Conclusion*, so that the section now summarizes the key findings, their implications for prebiotic chemistry, and perspectives for future work.

“**Conclusions**”

This study demonstrates that abyssal hydrothermal chimneys from multiple vent fields along the Indian Ocean ridge share the most common molecular connection patterns, regardless of their site or venting activity. Through systematic molecular fingerprinting, we reveal consistent molecular patterns supporting that accumulated hydrothermal processes actively drive the transformation from simple hydrocarbons to more complex and functionally enriched organic structures. By applying a co-annotation-based ‘phylogenetic’ framework, we reveal that these shared molecular features reflect the universal geochemical forcing of vent systems. Organic compounds preserved in these chalcopyrite-containing inner conduits record strong hydrothermal alteration and transformation.

Beyond that, these results provide new insights into the preservation and transformation of organic compounds under extreme submarine conditions, and they also establish a robust geochemical foundation for understanding how hydrothermal processes shape organic complexity, highlight potential analogues for prebiotic chemistry in early Earth hydrothermal systems, enabling imagination of origin-of-life scenarios. The shared molecular signatures may also offers a new framework for identifying molecular biosignatures of decreased or viable life in astrobiological contexts, e.g., the hydrothermal systems on early Mars and other

ocean worlds such as Enceladus and Europa.”

E. Suggested improvements: experiments, data for possible revision

According to the information provided, just the solvent extractable fraction was analysed. Is there any reason for this? Have you ever considered that another release method (the use of strong bases) might be useful? If possible, please essay this treatment, it possibly could give more data.

■ This is a good question. But we have clarify this very clearly.

We intentionally analyzed the solvent-extractable fraction for reasons. Our goal is to capture the pristine, least-altered, rock-hosted organics to characterize their molecular similarity.

Indeed, our choice of solvent extraction (1:1 dichloromethane:methanol) is based on experience in trace organic matter extraction from ultra-low TOC mineral-rich geological samples such as hydrothermal vent chimneys. In such samples, organic matter is extremely limited (S Table 1, 0.088 average), and any aggressive treatment, such as strong acids or bases being destructive, risks altering the molecular structures thereby creating artifacts, especially of fragile and diagnostic compounds (e.g., ester, carboxyls, amides, aromatic structures, other functional groups), which are central to our interpretations. Therefore, we deliberately avoided any potentially damaging extraction methods.

We would like to operate lesser and extract better. Solvent extraction represents a non-destructive method that ensures the preservation of the most pristine, indigenous organic molecules trapped in the rocks. Compared to other extraction techniques, it allows for the broadest and cleanest recovery of molecular classes, from small alkanes, to complex aromatics, within a single run. Other methods such as base hydrolysis are known to generate degradation products or artificial moieties, which would obscure the primary molecular information and hinder interpretation of in-situ chemical evolution.

We also note that we tested various DCM:MeOH ratios during method development. The 1:1 ratio used in this study provided the widest molecular coverage (the widest molecular spectrum from alkanes, to non-polar hydrocarbons to representative polar NSO species), whereas other conditions (e.g., 9:1) were less balanced for this work. Therefore, we consider the selected protocol to be a deliberate and optimal strategy ("zero-damage") for capturing the most representative molecular spectrum possible in such a rare and challenging sample type.

We agree that base-released fractions can be informative for soil/humic contexts, but in hydrothermal sulfide chimneys they are not appropriate for our objective of resolving common, native hydrothermal transformation patterns.

Relevant phrasings in text were revised or enhanced.

Even if the protocol for the extraction of organics is described in the methods section, the amount of each sample treated is not explicit nor the volume of solvents employed. In addition, it is not said if replicas were made for each analysis (Hg isotope, CG-MS, FT-ICR MS). Please clarify the previous point.

We have now clarified these details in the revised Methods section. Please refer to our previous response to comment “C” regarding our statistical procedures. Additional information on sample mass, replication, etc., for each analytical technique (Hg isotopes, GC-MS, FT-ICR MS) has been explicitly included to enhance clarity.

Specifically, the mother liquor concentration was 10 mg/mL in toluene, and injection was performed after dilution to 0.2 mg/mL using a toluene:methanol (1:3) mixture. The injection rate was 180 μ L/h up to 5 minutes.

The procedure for the FT-ICR MS is well described. However, I missed a detailed description of the samples analysed by FT-ICR MS. How much extract was analysed? Did you estimate before the concentration of organics in samples? Why did you select ESI and not API for study your samples? I understand that ESI is better for the analysis of polar and high molecular weight molecules. Nonetheless, many biomarkers are lipids and consequently, some of them are non-polar or slightly polar, so probably you could find nice information by using API. I guess if you performed the two analysis and chose the ESI instead of API for getting better results? If that is the case, please include the information. If not, it could be interesting to include those analyses in your samples.

■ We thank the reviewer for these constructive suggestions.

We clarify that FT-ICR MS was performed on the rest of solvent extracts rather than on the additional samples. This has now been clearly stated in the Methods section. Specifically, the mother liquor concentration was 10 mg/mL in toluene, and injection was performed after dilution to 0.2 mg/mL using a toluene:methanol (1:3) mixture. The injection rate was 180 μ L/h up to 5 minutes.

In our study, the organic content of the chimney samples is extremely low (TOC < ~0.1 mg/g), which severely limits the feasible ionization modes. After GCMS and prior to FT-ICR MS analysis, extracts were concentrated as much as possible, but the overall amount of soluble organics available was insufficient to reach the concentrations typically required for APPI (~0.5 mg/mL). Indeed, we performed test runs with APPI but found that the signal intensity was negligible compared with ESI.

Furthermore, given the low TOC and relatively high sulfur content in the total liquid extract (TLE), extensive desulfurization could cause excessive compound loss and introduce contamination, while methylation was also deemed unsuitable due to potential compositional biases. Direct use of APPI showed strong matrix interference

and poor ionization behavior, producing weak signals for biomarkers etc. and demanding higher concentrations than our samples could support. Therefore, ESI was selected as the optimal ionization method based on the continuity, safety, and integrity of the experimental and analytical workflow.

Based on our analytical experiences, the concentration requirements differ among ionization modes (methylated positive-ion ESI > APPI > positive/negative-ion ESI), with standard ESI being the most sensitive (requiring the lowest concentration). Given the very low TOC contents, ESI is better. Therefore, we focused on ESI, which is more sensitive under low-concentration conditions and particularly suitable for polar and heteroatom-bearing compounds.

In addition, positive-mode ESI efficiently ionizes basic species (or sodium-adducts of neutrals), whereas negative-mode ESI is more sensitive to acidic compounds and neutral nitrogen species. The latter allowed us to cross-validate our previous molecular phylogenetic tree results, where O- and N-bearing compounds appeared enriched in the later evolutionary stages. In particular, negative-ion ESI provides improved detection of O- and N-bearing compounds, which are of geochemical relevance to our study. For these reasons, negative-ion ESI was chosen as the optimal and most feasible ionization method for these extracts.

We have clarified this in the Methods section and included additional information in the Supplementary Tables regarding the sample amounts and ionization tests.

F. References: appropriate credit to previous work?

The references are appropriated, and the article is well documented, there is a bunch of articles in the HV area and authors selected key documents. I would just suggest including some references of the application of FT-ICR MS in other studies to trace the reach and possible meaning. For example:

- Butturini, A., Amalfitano, S., Herzsprung, P., Lechtenfeld, O. J., Venturi, S., Olaka, L. A., ... & Fazi, S. (2020). Dissolved organic matter in continental hydro-geothermal systems: insights from two hot springs of the East African Rift Valley. *Water*, 12(12), 3512.
- Ventura, G. T., Rossel, P. E., Simoneit, B. R., & Dittmar, T. (2020). Fourier transform ion cyclotron resonance mass spectrometric analysis of NSO-compounds generated in hydrothermally altered sediments from the Escanaba Trough, northeastern Pacific Ocean. *Organic Geochemistry*, 149, 104085.
- Gomez-Saez, G. V., Niggemann, J., Dittmar, T., Pohlabein, A. M., Lang, S. Q., Noowong, A., ... & Bühring, S. I. (2016). Molecular evidence for abiotic sulfurization of dissolved organic matter in marine shallow hydrothermal systems. *Geochimica et Cosmochimica Acta*, 190, 35-52.

Thanks for the suggestions.

We have reviewed these references, and added them to the relevant locations.

G. Clarity and context: lucidity of abstract/summary, appropriateness of abstract, introduction and conclusions

The manuscript is well written and there is a subdivision considering the main topics. The abstract is concise and gives information of the key findings of the research. However, there is not an introduction nor conclusion sections. Authors present a section called Broader significance at the end of the Discussion. However, I consider that they could include a final section (Conclusion) comprising the relevance of their data, the implications for prebiotic chemistry, their future work, etc.

Thank you for the valuable suggestion. In response, we have replaced the “Broader significance” section with a newly written concluding paragraph that explicitly addresses the relevance of our results, their implications for prebiotic chemistry, and directions for future work.

“Conclusions

This study demonstrates that abyssal hydrothermal chimneys from multiple vent fields along the Indian Ocean ridge share the most common molecular connection patterns, regardless of their site or venting activity. Through systematic molecular fingerprinting, we reveal consistent molecular patterns supporting that accumulated hydrothermal processes actively drive the transformation from simple hydrocarbons to more complex and functionally enriched organic structures. By applying a co-annotation-based ‘phylogenetic’ framework, we reveal that these shared molecular features reflect the universal geochemical forcing of vent systems. Organic compounds preserved in these chalcopyrite-containing inner conduits record strong hydrothermal alteration and transformation.

Beyond that, these results provide new insights into the preservation and transformation of organic compounds under extreme submarine conditions, and they also establish a robust geochemical foundation for understanding how hydrothermal processes shape organic complexity, highlight potential analogues for prebiotic chemistry in early Earth hydrothermal systems, enabling imagination of origin-of-life scenarios. The shared molecular signatures may also offers a new framework for identifying molecular biosignatures of decreased or viable life in astrobiological contexts, e.g., the hydrothermal systems on early Mars and other ocean worlds such as Enceladus and Europa.”

Further Questions

I consider very appealing the idea of tracing the evolution of molecules in HV;

however, it must be considered with caution. Taxonomy (hierarchical classification) is not the same as Phylogenetics (study of the evolutionary history) or Genomics (the study of structure, function, evolution of the genomes). You state that “For example, genomics supplies a static hereditary framework, yet hydrothermal alteration and transformation of organic compounds present a dynamic scaffold of geochemical information”. In fact, the genomes are constantly evolving, and evolutionary hypotheses (phylogenetic trees) establish the historical relationship between organisms. Biologists never consider genetic information to be static.

■ We thank the reviewer for this important clarification. We apologize for the confusion this may have caused. Genetic information is definitely NOT static, this was a rash statement. It was used to compare the different mechanisms and timescales of change: inherited genomic evolution versus the rapid, direct physicochemical transformations of organic molecules in the intense vent environment. The original wording was flawed. To address this, we have now very carefully revised and cleared relevant phrasings and clarified our conceptual framework.

We fully agree that our original phrasing lacked rigor in distinguishing between taxonomy, phylogenetics, and genomics. The taxonomy, phylogenetics, and genomics are distinct concepts. Our intention was not to equate hydrothermally altered molecular patterns with biological evolutionary frameworks. Rather, we tried to draw an analogy between molecular transformation trajectories and hierarchical clustering of molecular compositions more in the spirit of metabolomics than biological phylogenetics. Our idea is to track systematic changes in molecular profiles to reveal hydrothermal “metabolic” pathways. The term “evolution” in our context refers to the inferred progression (a presumed link) of organic geochemical alteration and complexification, not biological descent.

In the revision we have removed the wording that could be read as implying a “static” genomic framework. This is an inaccurate oversimplification and genomes are indeed dynamic and constantly evolving. We now present our approach as a metabolomics-inspired, network-based analysis of molecular relatedness in geochemical systems. We explicitly clarify that our organic geochemical “phylogenetic” tree is an analogy for a hierarchical, similarity-based relatedness tree built from molecular fingerprints, not a reconstruction of biological ancestry. The revised description more accurately reflects that our analysis provides geochemically-driven molecular continuity patterns under hydrothermal transformation, without implying direct biological analogy. These changes are made to avoid conceptual over-extending and improve clarity for readers.

“In *Results and discussion*.....section leading para

.....Once the spectra are purified (Supplementary Fig. 2), all subsequent calculations (pairwise cosine similarity, distance matrices, hierarchical clustering, and geochemical ‘phylogenetic’ reconstruction; Here, ‘phylogenetic’ is used by analogy to denote similarity-based hierarchical organization of molecular fingerprints, not a biological phylogeny of organisms) rely on well-established,

deterministic chemoinformatics and statistical routines.....”

“In *Network-informed molecular framework*.....section

.....For example, genomics provides a heritable framework through DNA sequences that record biological variation through time, and in an analogous way hydrothermal alteration and transformation of organic compounds provide a dynamic, process-based record of organic geochemical change driven by the hydrothermal environment. Rather than invoking biological phylogenetics, we adopt a perspective inspired by metabolomics, clustering chemically transformed compounds to explore their structural relationships and potential transformation pathways under environmental stress. Because these compounds possess markedly greater thermal stability, they preserve the dynamic imprint or gradient of hydrothermal alteration and transformation, offering a direct record of the “crucible” that vents provide for prebiotic molecular complexity.....

.....The network-based approaches, originally developed for studying systematic molecular change in living systems², are adopted here to non-enzymatic hydrothermal systems, where thermodynamic constraints govern transformation networks.....”

“In *Common evolution pathway from*.....section

.....We construct an organic geochemical relatedness tree, a network-based representation of molecular relationships inferred from untargeted mass-spectral fingerprints¹, analogous to phylogenetic tree for metabolites¹ or genomic sequences³. This usage is phylogeny-inspired for organizing molecular similarity, and does not imply biological ancestry.....”

Etc.

We are grateful to the reviewer for helping us refine our language. We believe these revisions now more accurately frame our methodology and prevent potential misunderstanding by a broader scientific audience, and underscore the unique contributions of organic geochemistry to understanding molecular complexity in extreme environments.

You mentioned that it is important to understand the mechanisms that allowed the formation and self-assembly of organic molecules under changeable conditions, including chemical and physical variables (i.e. pH, temperature, chemical composition, redox state, etc.). This is one of the most important questions in the study of hydrothermal vents as niches for prebiotic evolution. However, I do not find in your text a reflection about this, or the incorporation of some of those variables in the analysis of your data. In this sense, it could be important to have all the data of the sampling sites (Supp. Table 1); please, complete the information.

We agree with the reviewer that physicochemical variables (e.g., pH, temperature, fluid chemistry, redox state) are important to understand the self-assembly of organics

in hydrothermal systems. In our manuscript, we mentioned these factors as part of the theoretical overall background, and our current study was designed to focus on the conserved and shared compositional characteristics and statistical commonalities of organic molecules preserved in sulfide-hosted chimney structures, rather than directly modeling the effects of physical-chemical variables, despite the inter-site differences is also an interest framework. We agree that some descriptions in our manuscript may have overstated the mechanism interpretation. We have now revised the relevant statements to tone down such statements, and instead emphasize the role of our findings as observational constraints on hydrothermally modified organic matter, and provided more site and sample information in Supplementary Table 1, as far as available from that cruise, , to provide more context for readers. The water chemical parameters such as pH and temperature, rely on in situ real-time measurements, which are partly available but not complete.

We outline in the revised version that future work will explicitly integrate environmental variables with molecular change patterns, through more controlled sampling and micro-environmental data collection.

“This observed evolution pathway from chained/branched alkanes through aromatics to heteroatom-rich species, particularly amides, reflects a progressive molecular “heteroatomization” trajectory that is plausibly driven by the combined thermal and geochemical influence within hydrothermal vent systems. The shared compositional patterns observed in heterogeneous samples from vents of different activity levels and regions are best understood as aggregate responses to multiple environmental variables (e.g., heat, O₂, H₂, pH, redox potential, metals), rather than the effect of any single factor, that our current dataset cannot isolate the specific contributions of individual variables. In this sense, the integrated pattern provides observational constraints on how hydrothermal environments collectively shape organic molecular evolution, offering insights into the functional and structural transformation and dynamic metastability of organic molecules in abyssal hydrothermal contexts. Future work involving more resolved geochemical measurements and controlled sampling strategies will be essential to disentangle the relative influence of key environmental drivers.”

“.....The interplay between these opposing reaction trajectories may give rise to a complex transition network that is likely influenced by the overall physicochemical conditions of hydrothermal systems

Hydrothermal processes —including temperature, pressure, chemical gradients, Eh, pH, etc. — are responsible for greatly altering the organic molecules found in these environments. In your experience, how could we differentiate between organic molecules formed from simple precursors in HV and those resulting from the decomposition of complex organic matter from organisms? The remains of different organisms could be reaching the ocean floor through different means, even though

these organisms do not live near HV environments. The question relies on the fact that, if we could distinguish between those processes we could reinforce your hypothesis of molecular evolution.

■ Thank you for this insightful question which is also the most frequently asked question. We have carefully clarified and enhance this which is also an important context for our study.

“Although the precursor organic matter in this scenario should include biotic inputs, derived from microbial communities indigenous to or entrained within the hydrothermal system, the intense hydrothermal conditions have largely erased primary biological signatures. Instead, what remains is a set of highly transformed organic compounds that no longer retain bio-diagnostic features, but instead record the thermochemical legacy of the vent environment. In this context, cycloalkanes, polycyclic aromatic hydrocarbons (PAHs) and thiophenes, etc., for example, persist not as diagnostic biomarkers but as chemically inert feedstock continually reworked in the hydrothermal crucible, now essentially divorced from their original biological origins and transformed into molecular manifestations of hydrothermal processing. Vent samples from different regions with varying activity levels along the Indian Ridge could in principle have contained their own total molecular composition, however they most likely share certain proportions of common molecular geochemical characteristics. Thus those presumed shared characteristics and geochemical phylogenetic tree can provide essential constraints and critical contextual information for investigating mechanisms controlling organic molecular structural evolution and their metastable equilibrium states in abyssal extreme hydrothermal environments⁴.”

We agree that distinguishing between organic molecules formed abiotically from simple precursors and those derived from the degradation of complex biological macromolecules remains a challenge in organic geochemistry, particularly in hydrothermal vent (HV) settings.

In our experience, once exposed to $>300^{\circ}\text{C}$ high temperature intense hydrothermal conditions during chimney formation, most primary biological information in organic molecules is largely homogenized or demolished and ‘reset’, regardless of its fundamental origin. What we are able to extract and detect in core part of chimney are typically highly transformed products rather than pristine biosignatures or biomarker patterns (although there are still some but they contain no bio-diagnostic features, as they have been extensively reworked). This is consistent with observations from both highly thermal stress organic-rich rocks and epithermal ore deposits^{5,6}, where the original biological signatures are almost completely destroyed, and strongly overprinted compounds remain (see a figure below). This alteration and transformation render the origin of biotic or abiotic, which makes it is impossible to distinguish abiotic ones from this complex organic background.

Fig. Summed m/z (178 + 184 + 202 + 204 + 228 + 234 + 252 + 276 + 300) partial mass chromatograms; Identification of 1-phenyldibenzofuran and triphenyleno[1,12-bcd]furan in partial mass chromatograms (m/z 242 and 244)⁵

From this perspective, the source of the initial organic matter (biotic vs. abiotic) becomes less critical than its role as a common “organic substrate” for the natural laboratory of the vent system, and what we actually analyze represents the outcome of hydrothermal alteration rather than a direct fingerprint of its origin.

To the reviewer’s question of “how could we differentiate,” to our knowledge, our position is that no established method currently exists to unambiguously distinguish between biotic and abiotic sources within such a complex, heavily altered organic backgrounds. Tools like bulk carbon isotopes have also proven insufficient in this context⁷⁻⁹.

Indeed, evidence of abiotic hydrocarbons in natural earth systems is largely limited to the smallest molecules (methane, ethane, etc.)^{7,8,10-13}, whereas experimental high P-T synthesis studies (e.g., Ho-Kwang Mao, Haiyan Zheng and colleagues)^{10,14-16} and hydrothermal synthesis may have demonstrated that longer-chain hydrocarbons can be produced under laboratory conditions⁹. However, in natural hydrothermal systems, complex abiotic macromolecules remain unverified. Thus, the compounds we observe in hydrothermal chimneys are best regarded as products of ongoing alteration and transformation or molecular manifestations of hydrothermal processes, under specific vent conditions.

Therefore, our study does not simulate a laboratory-based stepwise synthesis from small to complex molecules, and we do not claim a trajectory directly and fully from abiotic precursors, we try to suggest a possible linkage between simple and functionalized organics shaped by natural hydrothermal conditions. Our study was designed not to trace the untraceable origin of every molecule, but to characterize the net result of the vent’s transformative power. The molecular assemblages we report are not primary signatures, but rather secondary altered products that record the processes of hydrothermal alteration and molecular evolution.

In this case, unlike controlled laboratory synthesis, which starts from pure inorganic precursors, submarine hydrothermal vents function as natural laboratories, where existing organic substrates from both biotic and abiotic precursors, regardless of

origin, are continually reworked, transformed, functionalized, and diversified, to largest extent. Our data contribute important constraints and natural analogs for this process.

In this context, the common molecular patterns we identified likely result from shared geochemical controls within HV systems, rather than from the inheritance of original biological structures.

While this work provides a framework for understanding their role as plausible settings for prebiotic chemistry, we acknowledge that definitively separating these sources, and the origin of life research, remain an open and unresolved challenge.

Hydrothermal vents during the Hadean Eon must have been shallower than contemporary submarine vents. Could you explain how we can determine the similarity between these systems?

■ Great question! Thank you for this one.

Indeed, hydrothermal systems in the Hadean Eon are presumed to be located at shallower settings compared to modern deep sea vents. However, from a geochemical and thermodynamic perspective, it is the thermochemical conditions rather than depth per se created by hydrothermal circulation that govern the transformation and synthesis of organic molecules, which can occur regardless of whether vents are shallow or deep.

Nonetheless, we have now toned down this wording and phrasings as ‘*early Earth hydrothermal systems*’ in the *Abstract* and new *Conclusion* section, to avoid confusion.

“.....The observed molecular evolution from simple alkanes to complex heteroatom-bearing compounds particularly nitrogenous species accumulating during vent shutdown, reveals a molecular “heteroatomization” trajectory that signals increasing functionalization and polarity in early Earth hydrothermal systems.....”

“.....highlight potential analogues for prebiotic chemistry in early Earth hydrothermal systems.....”

Our study does not aim to literally reconstruct Hadean environments directly, but rather to use modern submarine hydrothermal systems as a natural analogue that may mimic such extreme conditions on early Earth. A comparison to present-day hydrothermal systems should be understood as a process-based analogy rather than a literal depth equivalence. Contemporary vents provide observable examples of how simple organic molecules are transformed, functionalized under hydrothermal conditions. By extension, analogously underlying thermochemical processes could have occurred in Hadean shallow water systems, even if the physical settings differed.

Our interpretation therefore uses deep-sea hydrothermal vents as an analogue to

explore plausible prebiotic chemical pathways, and offering a plausible analogy for the molecular complexification that could have occurred in similar energetic settings on early Earth.

Figure 2 represents the distribution of organic compounds in a representative sample (SY139-G07; Edmond). Please, include another sample from an active site in the same figure. It could be also interesting to have the other data in Supplementary Content.

Sure, we included an active site sample. Other data are provided.

This shows the high hydrothermal alteration level of organic materials.

Minor comments

Regarding the data obtained from FT-ICR MS, please include the list of all identified molecules in the supplementary material (probably in an Excel spreadsheet).

Ok, this list of molecules have now been uploaded as supplementary files (within the relevant spreadsheets to Fig. 4)

 a - Longqi SY105 Active.xlsx

 b - Longqi SY093-G07 Inactive.xlsx

 c - Edmond SY150-G6 Inactive.xlsx

 d - Kairei SY145-G-1 Inactive.xlsx

Please have a check.

The supplementary material must include all the data presented in Figures, both in the main text or in the supplementary information; please incorporate this information.

All relevant data and spreadsheets have now been provided in the SI

Please check the text and correct the word earth (as a planet), replace it with Earth.

Corrected.

Line 378. Please give a temperature range in the sentence “significant high-temperature activity”.

A rough temperature range was provided: (>~300°C).

Line 409. Remove the point after “(e.g., cysteine) 4,8,19,51,52., and”.

Thanks. This part was removed.

Line 452- Correct the subindex in (HNO₃).

Corrected.

List of changes response to Reviewer #2

Reviewer #2 (Remarks to the Author):

The abstract and paper needs major rewrite for more general science audience.

Rewritten abstract as follows

Abyssal hydrothermal vents are regarded as crucibles for prebiotic organic chemistry. While genomics and metabolomics have been extensively studied in such settings, the organic geochemical continuum and evolutionary transitions that link simple abiotic reduced carbon to heteroatom-rich polymer precursors remain poorly constrained, largely due to the highly dynamic hydrothermal alteration and transformation. Here, we apply a metabolomics-inspired strategy incorporating both hard and soft ionization, algorithmic deconvolution of mass spectra, high-precision structural identification and molecular similarity matching, and hierarchical organization, to construct a molecular-relatedness organic geochemical phylogenetic tree of vents from ultraslow-spreading Southwest Indian Ridge. We report the shared evolutionary relationship among organic molecules across different vent sites of differing geologic activity.

Molecular progressive reconstruction of alkanes, alkanols, phenyls, PAHs, NSO-compounds, suggests that, regardless of abiotic or biotic origin, these molecules have been extensively transformed under hydrothermal conditions. The observed molecular evolution from simple alkanes to complex heteroatom bearing compounds particularly nitrogenous species accumulating during vent shutdown- reveals a “heteroatomization” trajectory that signals increasing functionalization and polarity in the Hadean ocean. This finding helps bridge the gap between abiotic reduced carbon compounds and life’s essential organic feedstock, further underscoring the role of hydrothermal systems in mediating the transition from abiotic molecules to prebiotic chemistry. It provides a foundational pathway for the emergence of life’s building blocks on the primordial Earth

Thank you for your help with the clarity and simplicity of the abstract.

We accepted your suggestions and carefully revised the abstract. It looks snappier now!

New abstract below:

“Abyssal hydrothermal vents are regarded as crucibles for prebiotic organic chemistry. While genomics and metabolomics have been extensively studied in such settings, the organic geochemical continuum and evolutionary transitions that link simple abiotic reduced carbon to heteroatom-rich polymer precursors remain

poorly constrained, largely due to the highly dynamic hydrothermal alteration and transformation. In this work, we apply a metabolomics-inspired strategy incorporating both hard and soft ionization, deconvolution of mass spectra, high-precision structural identification and molecular similarity matching, and hierarchical organization, to construct a molecular-relatedness organic geochemical ‘phylogenetic’ tree of vents from ultraslow-spreading Southwest Indian Ridge. Here, we report the shared evolutionary relationship among organic molecules across different vent sites with differing activity levels. Molecular progressive reconstruction of alkanes, alkanols, phenyls, PAHs, NSO-compounds, suggests that, regardless of abiotic or biotic origin, these molecules have been extensively transformed under hydrothermal conditions. The observed molecular evolution from simple alkanes to complex heteroatom-bearing compounds particularly nitrogenous species accumulating during vent shutdown, reveals a molecular “heteroatomization” trajectory that signals increasing functionalization and polarity in early Earth hydrothermal systems. This finding helps bridge the gap between abiotic reduced carbon compounds and life’s essential organic feedstock, further underscoring the role of hydrothermal systems in mediating the transition from abiotic molecules to prebiotic chemistry. It provides a foundational pathway for the emergence of life’s building blocks on the primordial Earth.”

I found this paper very difficult to follow- the overall aims needs carefully articulating.

This study investigates the molecular evolution of organic compounds in abyssal hydrothermal vents, environments thought to be key sites for prebiotic chemistry. Despite well-documented genomic and metabolomic studies in such settings, the transition from simple abiotic molecules to complex prebiotic organics remains poorly understood due to intense hydrothermal alteration. Using a metabolomics-inspired strategy—including advanced mass spectrometry, algorithmic spectral deconvolution, and molecular similarity mapping—the authors constructed a geochemical phylogenetic tree from vents along the ultraslow-spreading Southwest Indian Ridge. They identified evolutionary links among diverse organic molecules (e.g., alkanes, alcohols, polycyclic aromatics, and nitrogen/sulfur/oxygen-containing compounds), revealing a continuum from simple hydrocarbons to more complex, heteroatom-rich structures. These transformations occur irrespective of the molecules’ biotic or abiotic origins, with shutdown phases of venting showing an increased presence of nitrogenous compounds. This progression reflects a molecular “heteroatomization” pathway, suggesting increasing functionality and polarity—key traits for life’s chemical precursors. The findings highlight hydrothermal systems as crucibles for the stepwise evolution of reduced carbon toward biologically relevant molecules, helping to bridge the gap between abiotic chemistry and the emergence of prebiotic organics on early Earth.

We thank the reviewer for highlighting this concern. We agree that clear articulation of the overarching aims is essential for guiding readers through the study. We have carefully revised the framework of this MS to enhance readability for wider audience.

In the revised version, we have refined the *Introduction* to better highlight the motivation and objectives of our work, and to explicitly state the central aims of our study in a more focused and structured way.

“.....

By building this “tree” on both active and inactive hydrothermal vents from Longqi, Edmond, and Kairei vent sites along the Indian Ridge (Supplementary Fig. 1), the specific aim of this study is to determine whether hydrothermal processes impose a shared molecular trajectory from simple alkanes to complex, functionalized N-, S- and O-bearing molecules, and to visualize the pathways of hydrothermal organic alterations and transformations in such systems⁶, and to explore their implications for prebiotic chemistry in early Earth environments. This work may offer insights into the evolution of life-related substances in primordial ocean world, and this framework may also contribute to the search for decreased life markers on Mars providing a conceptual model for identifying biosignatures in astrobiological contexts.”

We now clarify that our primary goal is to identify and characterize common molecular features in hydrothermal sulfide chimney samples, which can shed light on the abiotic to prebiotic transition under hydrothermal conditions.

We emphasize that the study is not focused on comparing different hydrothermal fields per se, but rather on extracting shared molecular patterns across diverse chimney samples, to explore the common geochemical pathways that may drive prebiotic molecular evolution.

We have also recreated a final *Conclusion* to explicitly state the significance of our findings in relation to prebiotic chemistry.

“Conclusions

This study demonstrates that abyssal hydrothermal chimneys from multiple vent fields along the Indian Ocean ridge share the most common molecular connection patterns, regardless of their site or venting activity. Through systematic molecular fingerprinting, we reveal consistent molecular patterns supporting that accumulated hydrothermal processes actively drive the transformation from simple hydrocarbons to more complex and functionally enriched organic structures. By applying a co-annotation-based ‘phylogenetic’ framework, we reveal that these shared molecular features reflect the universal geochemical forcing of vent systems. Organic compounds preserved in these chalcopyrite-containing inner conduits record strong hydrothermal alteration and transformation.

Beyond that, these results provide new insights into the preservation and transformation of organic compounds under extreme submarine conditions, and

they also establish a robust geochemical foundation for understanding how hydrothermal processes shape organic complexity, highlight potential analogues for prebiotic chemistry in early Earth hydrothermal systems, enabling imagination of origin-of-life scenarios. The shared molecular signatures may also offers a new framework for identifying molecular biosignatures of decreased or viable life in astrobiological contexts, e.g., the hydrothermal systems on early Mars and other ocean worlds such as Enceladus and Europa.”

We believe these revisions now make the aims of the paper much clearer and easier to follow.

It is very difficult to assess the quality of data generated from the samples investigated. The input data needs to be vigorously validated - from the origin of the samples, preparation of samples, procedural blanks through the whole process needs careful consideration. The samples cover a range of compounds -polarity and some of these compounds are likely introduced contaminants from storage-to analyses. This is my biggest concern with the data. More background information is needed before this can be published.

We appreciate the reviewer’s concern regarding the quality and reliability of the input data. We agree that, for samples of such rarity and complexity, rigorous validation and contamination control are of great importance. We acknowledge that some of these details may not have been sufficiently clear in the initial manuscript. To improve clarity and accessibility of our study, we have substantially revised the manuscript by expanding on the following aspects:

Sample origin: We have now included a detailed description of the *Sampling description*, including comprehensive background information on the research cruises, tools (e.g., HOV *Shenhai Yongshi*), vent locations, and sampling rationale, etc. (see Methods section and Supp. Table 1). The sample origin is now clearly established based on controlled deep-sea expeditions.

Sample preparation and contamination control: Based on our multi experiences of wet-chemistry extraction in organic geochemistry field^{5,6,17}, we now explicitly describe contamination minimization and pre-treatment steps applied from retrieval to organic extraction, including materials, labware cleaning, and solvent systems in each method sub-sections. We also clarified that standards (AccuStandard, Inc), procedural blanks and burning blanks were run alongside rare samples to ensure analytical confidence, common laboratory contaminants, such as siloxanes, were identified during data processing and have been excluded from our final datasets, and replicate checks were performed where possible (see revised Methods and Supp. Table 2).

Analytical strategy: Given the extremely low TOC contents and limited available material, we carefully optimized extraction and analysis protocols (e.g., 1:1 DCM:MeOH based on expected wider molecular spectrum of compound classes from one operation). We also clarified that our approach was designed to maximize the

information obtainable, while minimizing any destructive treatments. This included not only optimizing solvent extraction protocols but also carefully selecting complementary ionization modes.

Balance between detail and readability: In line with Nature Communications editorial guidelines, we ensured that the Methods section remains concise yet comprehensive, providing sufficient information for interpretation and replication of results.

Additional technical specifics and references have been included in supplementary materials or cited accordingly to enhance transparency and reproducibility while maintaining clarity for a broad readership.

We now have made a significant effort to ensure data quality, clarity, and reliability. We hope the expanded descriptions address the reviewer's concerns and clarify that the data are derived from robust procedures, with contamination control and analytical accuracy carefully considered throughout.

References in this response to the reviewers

- 1 Tripathi, A. *et al.* Chemically informed analyses of metabolomics mass spectrometry data with Qemistree. *Nat. Chem. Biol.* **17**, 146-151 (2021).
- 2 Aksenov, A. A. *et al.* Auto-deconvolution and molecular networking of gas chromatography–mass spectrometry data. *Nat. Biotechnol.* **39**, 169-173 (2021).
- 3 Davín, A. A. *et al.* A geological timescale for bacterial evolution and oxygen adaptation. *Science* **388**, eadp1853 (2025).
- 4 Shock, E. L. *et al.* Thermodynamics of organic transformations in hydrothermal fluids. *Rev. Mineral. Geochem.* **76**, 311–350 (2013).
- 5 Xu, H. *et al.* Hydrothermal catalytic conversion and metastable equilibrium of organic compounds in the Jinding Zn/Pb ore deposit. *Geochim. Cosmochim. Acta* **307**, 133–150 (2021).
- 6 Xu, H. *et al.* Organic compounds in geological hydrothermal systems: A critical review of molecular transformation and distribution. *Earth-Sci. Rev.* **252**, 104757 (2024).
- 7 Sherwood Lollar, B., Westgate, T. D., Ward, J. A., Slater, G. F. & Lacrampe-Couloume, G. Abiogenic formation of alkanes in the Earth's crust as a minor source for global hydrocarbon reservoirs. *Nature* **416**, 522–524 (2002).
- 8 Sherwood Lollar, B. Life's Chemical Kitchen. *Science* **304**, 972 (2004).
- 9 He, D. *et al.* Hydrothermal synthesis of long-chain hydrocarbons up to C₂₄ with NaHCO₃-assisted stabilizing cobalt. *Proc. Nat. Acad. Sci. U.S.A.* **118**, e2115059118 (2021).
- 10 Scott, H. P. *et al.* Generation of methane in the Earth's mantle: *In situ* high pressure–temperature measurements of carbonate reduction. *Proc. Nat. Acad. Sci. U.S.A.* **101**, 14023–14026 (2004).
- 11 Proskurowski, G. *et al.* Abiogenic hydrocarbon production at Lost City hydrothermal field. *Science* **319**, 604 (2008).
- 12 Klein, F., Grozeva, N. G. & Seewald, J. S. Abiotic methane synthesis and serpentinization in olivine-hosted fluid inclusions. *Proc. Nat. Acad. Sci. U.S.A.* **116**, 17666 (2019).
- 13 Seewald, J. S. Evidence for metastable equilibrium between hydrocarbons under hydrothermal

- conditions. *Nature* **370**, 285–287 (1994).
- 14 Yang, X. *et al.* Chemical transformations of n-hexane and cyclohexane under the upper mantle conditions. *Geosci. Front.* **12**, 1010–1017 (2021).
- 15 Tao, R. *et al.* Formation of abiotic hydrocarbon from reduction of carbonate in subduction zones: Constraints from petrological observation and experimental simulation. *Geochim. Cosmochim. Acta* **239**, 390–408 (2018).
- 16 Zhang, L. *et al.* Massive abiotic methane production in eclogite during cold subduction. *National Science Review* **10**, nwac207 (2023).
- 17 Xu, H. *et al.* Millimetre-scale biomarker heterogeneity in lacustrine shale identifies the nature of signal-averaging and demonstrates anaerobic respiration control on organic matter preservation and dolomitization. *Geochim. Cosmochim. Acta* **348**, 107–121 (2023).

We thank two anonymous reviewers and their helpful comments and constructive feedback. The list of changes and our **Point by Point Response** to the two reviewers can be found in the **blue text** below, and **Quotes** from our revised text in the manuscript are in **green**. All suggestions and opinions were fully considered and accepted. Corrections and revisions have been carefully made. Some comments and encouragement that were made will be of great help for our continued work, and are much appreciated. Please note that **the line numbers** and **the figures** in the manuscript have **greatly shifted** after revision, and the change of citations by **Endnote** cannot be tracked by **office word program**, and the changes in tables, figures and other materials may not be tracked either. We have submitted a copy of the *Revised Manuscript with Track Changes* and a **clean copy of the Manuscript file**. The *Supplementary Information* (a clean copy and one with Track Changes) were uploaded as well.

REVIEWER COMMENTS (NCOMMS-25-48088B)

List of changes response to Reviewer #1

Reviewer #1 (Remarks to the Author):

Dear authors and Editor,

I hope that you are doing well. I am sending to you my second review to the manuscript titled “Abyssal hydrothermal alteration drives the evolution from simple alkanes to prebiotic molecular complexity”, authored by Liu and coauthors.

The article proposes a hypothesis about the progression in the composition and structure of organic molecules, driven by likely natural processes in primitive environments. As an example, they propose the study of organics in HVS. The approach is novel and could certainly be a good topic for discussion in the scientific community.

In my previous comments, I raised some points that could improve the manuscript. The authors re-elaborated their manuscript and addressed all my earlier suggestions thoroughly. I greatly appreciate the time taken to respond and the in-depth answers.

Also, the required data and graphic material were properly provided, analysed and presented.

The authors explained the methods at detail, as well as the reasons that underpin their

decisions regarding sampling, analytical techniques, use of statistics, and data analysis. The result is a clearer and more robust manuscript. I very much acknowledge the inclusion of the photographs of samples and raw data; this is great for readers.

I have no more comments to the manuscript, just say that it represents a good contribution to the academic discussion.

Dear reviewer,

We sincerely thank you for the evaluation of our revised manuscript. Your thoughtful comments and suggestions have significantly improved the clarity, robustness, and credibility of our manuscript. The revisions based on your input have helped us to eliminate confusions and enhance the overall readability, transparency and coherence of the work. We deeply appreciate your time and constructive feedback.

List of changes response to Reviewer #2

Reviewer #2 (Remarks to the Author):

Unfortunately the manuscript has not satisfied concerns with regards to contamination. None of the procedural blanks and analyses have been included in the manuscript. This causes concerns on the origin of compounds and even more so when samples have extremely low TOC%.

Dear reviewer,

We sincerely thank you for your additional concerns regarding potential contamination. This issue is indeed of central importance for any organic geochemical investigation and explanation, particularly when the analyzed samples have low TOC contents. We have taken this opportunity to further clarify and visualize our contamination control measures and results, which helps us better clarify and validate the reliability of our dataset, and this also helps we understand our data better.

To address your concern, we have now added Supplementary Figs. 7–9, related descriptions in the revised Supplementary Information and associated metadata (i.e., raw data; deposited in Zenodo at

https://zenodo.org/records/17398264?preview=1&token=eyJhbGciOiJIUzUxMiJ9.eyJpZCI6ImZiYTcyYjNlTjZiZWYtNGYxNi00YTdmLTl2YWJhNWU3NmVhYSIsImRhdGEiOiOnt9LCJyYW5kb20iOiIwNzQxMzVlNTM1ZTU5ODFmNThlMmVhMjI4ZjNjNmYySj9xL3W2SeUdPAyLGQMTFEup9dx8tfQZ_mkVSUVW6x-Pe97ard-QOlecJpEMCZm1RxZyge5DXiC0VcSsxPONmnG_g), which together provide direct evidence of our contamination evaluation and the validity of the organic signals:

We have now included a comparison of airborne, burning, and procedural blank

samples (Supplementary Fig. 7), all of which show extremely low background signals (<0.1–2% of sample intensities; two orders of magnitude higher than any background). This supports that the strong organic signals in real samples are not from lab-based contamination. Supplementary Fig. 7 shows airborne and procedural blank tests conducted under laboratory conditions, revealing negligible organic signals compared to the samples tested.

Supplementary Fig. 7

Supplementary Fig. 8 illustrates a typical chromatographic appearance of hydrothermally reworked organic matter characterized by a strong unresolved complex mixture (UCM), contrasted against unaltered sedimentary rocks. As shown in our new Supplementary Fig. 8, hydrothermal alteration actively destroys primary, resolved biomarkers (like hopanes) and transforms them into a complex "hump" of altered molecules (the UCM), this UCM representing a rich reservoir of transformed compounds. This demonstrates that a UCM signal is an intrinsic product of hydrothermal reworking rather than an indication of a failed or contaminated analysis.

Supplementary Fig. 8

Despite low TOC contents in some samples, the compound-specific intensities from GC-MS and FTMS remain significantly higher than any blank, by two orders of magnitude. Supplementary Fig. 9 directly compares mass spectra from the procedural blank and from hydrothermal chimney extracts (SY105 and SY150-G6), and the blank spectra show extremely low intensity (~2% of that from sample tested). Peak profiles, compound richness, and molecular weight distributions show clear distinction from any known lab contamination, confirming that the detected organic compounds are not laboratory artifacts.

Supplementary Fig. 9

All potential contamination sources were systematically controlled and minimized. In our conventional processes, we employed fixed-vial systems within fume hoods to capture airborne organics, pre-cleaned all glassware and tools, used purified solvents, and cut outside and rinsed chimney interiors to isolate unexposed cores. These procedures follow well-established contamination control protocols in organic geochemistry (Brocks et al., 2008; Flannery and George, 2014; French et al., 2015; Brocks et al., 2017; Xu et al., 2023).

In total, 654 mass spectra were commonly annotated across all samples. Any possible laboratory contaminants (such as phthalates, siloxanes) were carefully filtered out during spectral similarity matching and scoring, which have no influence on the matrix construction or clustering process and topology of the organic geochemical relatedness tree. Importantly, our analysis does not aim at targeted compound identification or interpretation of individual compounds and their indication of biomolecular origins or environmental indicators or other proxies. Instead, it focuses on a more fundamental deeper layer of mass spectral data, i.e., the fragmentation behavior, the structural similarity patterns among deconvoluted mass spectra, which represents the true signal (metadata; actually those m/z number sequences from 42, 85, 191, 231, 245... etc.) of these analyses. By analyzing the architecture of

fragmentation-based spectral similarity, we aim to find molecular-level convergence patterns that may reflect the molecular processes within the hydrothermal system. This inspired strategy emphasizes structural similarity and fragmentation variations rather than the specific molecular identities, providing a framework to understand how organic assemblages evolve, transform and converge within abyssal hydrothermal systems.

Finally, we would also like to address the concern that "low TOC%" implies a weak signal susceptible to contamination. The prominent Unresolved Complex Mixture (UCM) in our chromatograms is, in fact, a key geochemical signal, not a weakness. We have clarified that the occurrence of UCM in our mass chromatograms is not indicative of low/incomplete resolution or analytical contamination, but reflects the extensive thermal and catalytic transformation of complex organics within hydrothermal systems, leading to abundant yet co-eluting fragments. One common source of misunderstanding comes from the presence of large unresolved complex mixture (UCM; the humped 'baseline') in our chromatograms. While often considered as "low-abundance" or "uninterpretable" signals in study field of environments and geography and some low-mature range organic petrology, UCMs are actually typical for thermally overprinted or hydrothermally altered organics (see Supplementary Fig. 8, (Xu et al., 2024)). These UCMs reflect secondary alteration and co-elution of structurally complex fragments, and are frequently seen in high temperature range geological settings, e.g., hydrothermal vents (e.g., Simoneit and Lonsdale, 1982; Simoneit, 2018).

We believe these additions and clarifications demonstrate that contamination was strictly controlled, procedural blanks were thoroughly examined, and the detected signals are representative of the hydrothermal samples, and the complex molecular signatures we report are indigenous and far exceeding any background. We hope these address your concerns and provide a clearer demonstration of our data robustness and experimental reliability.

We thank you again for your critical feedback, which has pushed us to present our data with greater clarity and rigor.

Sincerely,

Liu Quanyou, PhD; Huiyuan Xu, PhD

On behalf of the co-authors who have all approved this revision.

References in this response to the reviewers

- Brocks, J.J., Grosjean, E., Logan, G.A., 2008. Assessing biomarker syngeneity using branched alkanes with quaternary carbon (BAQCs) and other plastic contaminants. *Geochimica et Cosmochimica Acta* 72, 871–888.
- Brocks, J.J., Jarrett, A.J.M., Sirantoine, E., Hallmann, C., Hoshino, Y., Liyanage, T., 2017. The rise of algae in Cryogenian oceans and the emergence of animals. *Nature* 548, 578–581.

- Flannery, E.N., George, S.C., 2014. Assessing the syngeneity and indigeneity of hydrocarbons in the ~1.4 Ga Velkerri Formation, McArthur Basin, using slice experiments. *Organic Geochemistry* 77, 115–125.
- French, K.L., Hallmann, C., Hope, J.M., Schoon, P.L., Zumberge, J.A., Hoshino, Y., Peters, C.A., George, S.C., Love, G.D., Brocks, J.J., Buick, R., Summons, R.E., 2015. Reappraisal of hydrocarbon biomarkers in Archean rocks. *Proceedings of the National Academy of Sciences of the United States of America* 112, 5915–5920.
- Simoneit, B.R.T., Lonsdale, P.F., 1982. Hydrothermal petroleum in mineralized mounds at the seabed of Guaymas Basin. *Nature* 295, 198–202.
- Simoneit, B.R.T., 2018. Hydrothermal Petroleum, in: Wilkes, H. (Ed.), *Hydrocarbons, oils and lipids: Diversity, origin, chemistry and fate*. Springer, Berlin, pp. 1–35.
- Xu, H., Hou, D., Löhr, S.C., Liu, Q., Jin, Z., Shi, J., Liang, X., Niu, C., George, S.C., 2023. Millimetre-scale biomarker heterogeneity in lacustrine shale identifies the nature of signal-averaging and demonstrates anaerobic respiration control on organic matter preservation and dolomitization. *Geochimica et Cosmochimica Acta* 348, 107–121.
- Xu, H., Liu, Q., Jin, Z., Zhu, D., Meng, Q., Wu, X., Li, P., Zhu, B., 2024. Organic compounds in geological hydrothermal systems: A critical review of molecular transformation and distribution. *Earth-Science Reviews* 252, 104757.

We thank the anonymous reviewer 2 and the helpful comments and constructive feedback. The list of changes and our **Point by Point Response** to the two reviewers can be found in the **blue text** below, and **Quotes** from our revised text in the manuscript are in **green**. All suggestions and opinions were fully considered and accepted. Corrections and revisions have been carefully made.

Please note that **the line numbers** and **the figures** in the manuscript may have **greatly shifted** after revision, and the change of citations by **Endnote** cannot be tracked by **office word program**, and the changes in tables, figures and other materials may not be tracked either. We have submitted a copy of the **Revised Manuscript with Track Changes** and a **clean copy of the Manuscript file**. The **Supplementary Information** (*a clean copy and one with Track Changes*) were uploaded as well.

REVIEWER COMMENTS (NCOMMS-25-48088C)

List of changes response to Reviewer #2

Reviewer #2 (Remarks to the Author):

I have reviewed the authors' response regarding contamination and I am largely satisfied with the revisions. However, I still have some concerns that the sterols may have been introduced during the workup procedures. Sterols are common laboratory contaminants and can easily be absorbed onto samples. I would like the authors to be fully certain that the sterols reported here are genuinely indigenous.

We thank the reviewer for their satisfaction with our previous revision and for this important point. We conducted an additional round of evaluation focused on sterols (cholesterol, sitosterol, etc.), including re-checking all chromatograms, procedural blank, and other controls.

Our new Supplementary Fig. 10a compares the sterol-diagnostic ions in airborne blank, burning blank, procedural blank and chimney extracts. Possible sterol-related peaks (mass spectra) were absent at the retention times characteristic of sterols or their derivatives in all blanks, thus, sterols were not introduced during workup, indicating minimal to negligible background contamination. However chimney extracts show these indigenous mass spectral intensities an order of magnitude higher around $\times 10^3$, and we have to report this mass fragment pattern exactly as observed, while clarifying that they do not affect the conclusions of this study. We believe that this trace level signals should be reworked residues resulting from complex hydrothermal conditions. We slightly revise the Fig. 1 and relevant text in MS.

Published works reported sterols in submarine hydrothermal systems (Simoneit, 1990; Lattuati et al., 1998; Medeiros and Simoneit, 2007; Simoneit, 2008), supporting the possible indigenous nature of these compounds in hydrothermal environments. The presence of sterols in hydrothermal chimney matrices is geologically expected, although they are not used for the interpretive purpose in our study.

So we added clarification in the revised manuscript, trace sterols are mentioned only as part of the detected molecular inventory, they do not influence any evolutionary pathway, similarity analysis, or tree topology, and they are excluded from key interpretations. The presence of trace sterols does not affect any conclusion of the study, that hydrothermal alteration drives molecular evolution based on the broad, systematic transformation patterns observed across multiple compound classes.

Supplementary Fig. 10a Null detection of sterols in the procedural blank.

A continuum of related compounds has previously been observed in a Devonian crustacean fossil, where the entire continuum was attributed to microbial transformations. The authors should include a reference to this study (Melendez et al., 2013, Scientific Reports) and briefly discuss in the manuscript other processes that may account for a diagenetic continuum.

We appreciate the reviewer for this insightful comment. We have carefully evaluated Melendez et al., 2013 (Scientific Reports), and we incorporated this and other relevant citations (Melendez et al., 2013b; Melendez et al., 2013a; Lengger et al., 2017) and discussions in the revised MS. This additionally improves the scientific rigor of our work.

We now explicitly acknowledge that “molecular continuum” may arise from multiple processes, including, slow geological diagenesis, microbial transformation, thermal stress and cracking, and quick hydrothermal alteration (our focus).

The key distinction is that Melendez et al. studied an anoxic, buried Devonian crustacean fossil from a molecular taphonomic perspective over geological burial time ($\sim 10^8$ years) without a rapid thermal event, where e.g., slow microbial and diagenetic processes dominate the interpretations. This scenario is a high-temperature hydrothermal chimney forming on timescales of years to $\sim 10^4$ years (Simoneit, 2018), involving, 200–350 °C fluid, rapid mineralization, sulfide precipitation, strong catalytic surfaces, etc. The organic matter in our samples displays very low level and is embedded within sulfide minerals especially chalcopyrite (CuFeS_2), suggesting strong thermal degradation on whole organic materials rather than selectively microbial reworking. UCM is a strong indicator that can be readily created under hydrothermal conditions (Rushdi and Simoneit, 2011). Moreover, secondary biomarkers (Peters and Moldowan, 1991; Bennett et al., 2006) such as norhopanes diagnostic for subsequent microbial alteration were not observed (Supplementary Fig. 10b). So microbial transformation during diagenesis is not expected to be a dominant process in the interior of the chimney rocks, at temperatures (200–350 °C), where lipids crack rapidly, and proteins/peptides/nucleic acids cannot survive, and microbe viability is restricted to the adequate fringes at the steep thermal gradient zone.

Overall, we have added a paragraph discussing alternative processes, and we have clarified that hydrothermal thermochemical overprinting, not biological diagenesis, is the primary driver in this system, though microbial inputs certainly exist, and biological processes cannot be entirely excluded. We acknowledge that some structural convergence could arise from multiple processes and therefore avoid over interpretation. Some revision were made based on the Melendez example and we emphasized the unique context of hydrothermal chimneys (timescale, temperature) as distinct from diagenetic process, while noting the complementary nature of such studies.

Supplementary Fig. 10b

“A continuum of related compounds has been reported from fossilized crustacean and mineral deposits^{40,51}, where the entire series can be attributed to microbial transformation and sedimentary diagenesis and thermochemical alteration. We now acknowledge that such biological or diagenetic pathways can also produce molecular continua. However, the microbial mechanisms are unlikely to dominate in this scenario. The vent interiors examined here experience rapid hydrothermal

heating (>300 °C), extremely low TOC, and steep redox and temperature gradients, conditions under which microbial degradation is suppressed and thermochemical alteration overwhelmingly prevails. Unlike sedimentary diagenesis, these systems lack diagnostic biodegradation markers, and instead show UCM signatures and aromatic/heterocyclic enrichment characteristic of hydrothermal transformation along with the chalcopyrite, further supporting a non-diagenetic mechanism. Although microbial mechanism cannot be entirely excluded, the observed continuum is more plausibly attributed to rapid transformation in hydrothermal settings rather than slow geological diagenesis⁵¹⁻⁵⁴.”

References in this response to the reviewers

- Bennett, B., Fustic, M., Farrimond, P., Huang, H., Larter, S.R., 2006. 25-Norhopanes: Formation during biodegradation of petroleum in the subsurface. *Organic Geochemistry* 37, 787–797.
- Lattuati, A., Guezennec, J., Metzger, P., Largeau, C., 1998. Lipids of *Thermococcus hydrothermalis*, an archaea isolated from a deep-sea hydrothermal vent. *Lipids* 33, 319-326.
- Lengger, S.K., Melendez, I.M., Summons, R.E., Grice, K., 2017. Mudstones and embedded concretions show differences in lithology-related, but not source-related biomarker distributions. *Organic Geochemistry* 113, 67–74.
- Medeiros, P.M., Simoneit, B.R., 2007. Gas chromatography coupled to mass spectrometry for analyses of organic compounds and biomarkers as tracers for geological, environmental, and forensic research. *Journal of Separation Science* 30, 1516-1536.
- Melendez, I., Grice, K., Schwark, L., 2013a. Exceptional preservation of Palaeozoic steroids in a diagenetic continuum. *Scientific Reports* 3, 2768.
- Melendez, I., Grice, K., Trinajstić, K., Ladjavardi, M., Greenwood, P., Thompson, K., 2013b. Biomarkers reveal the role of photic zone euxinia in exceptional fossil preservation: An organic geochemical perspective. *Geology* 41, 123–126.
- Peters, K.E., Moldowan, J.M., 1991. Effects of source, thermal maturity, and biodegradation on the distribution and isomerization of homohopanes in petroleum. *Organic Geochemistry* 17, 47–61.
- Rushdi, A.I., Simoneit, B.R.T., 2011. Hydrothermal alteration of sedimentary organic matter in the presence and absence of hydrogen to tar then oil. *Fuel* 90, 1703–1716.
- Simoneit, B.R., 2008. Natural products as biomarker tracers in environmental and geological processes, Selected topics in the chemistry of natural products. *World Scientific*, pp. 77-126.
- Simoneit, B.R.T., 1990. Hydrocarbons associated with hydrothermal minerals, vent waters and talus on the East Pacific Rise and Mid-Atlantic Ridge. *Applied Geochemistry*.
- Simoneit, B.R.T., 2018. Hydrothermal Petroleum, in: Wilkes, H. (Ed.), *Hydrocarbons, oils and lipids: Diversity, origin, chemistry and fate*. Springer, Berlin, pp. 1–35.